# Assessing decadal to centennial scale nonstationary variability in meteorological drought trends

Kyungmin Sung[1,2], Max Torbenson[3], James H. Stagge[1],

[1]Civil, Environmental and Geodetics Engineering, The Ohio State University, Columbus, Ohio
[2]Adaptation Center for Climate Change, Korea Environment Institute, Sejong, Republic of Korea
[3]Department of Geography, Johannes Gutenberg Universität, Mainz, Germany

*Correspondence to*: Kyungmin Sung (sung.229@osu.edu)

**Abstract.** There are indications that the reference climatology underlying meteorological drought has shown non-stationarity at seasonal, decadal, and centennial time scales, impacting the calculation of drought indices and potentially having ecological and economic consequences. Analyzing these trends in the meteorological drought climatology beyond the 100-year, which exceeds available observation data period contributes to a better understanding of the non-stationary changes, ultimately determining whether they are within the range of natural variability or outside this range. To accomplish this, our study introduces a novel approach to incorporate unevenly scaled tree-ring proxy data (NASPA) with instrumental precipitation datasets by first temporal downscaling the proxy data to produce a regular time series, and then modeling climate non-stationarity while simultaneously correcting model induced bias. This new modeling approach was applied to 14 sites across the continental United States using the 3-month Standardized Precipitation Index (SPI) as a basis. Findings showed locations which have experienced recent rapid shifts towards drier or wetter conditions during the instrumental period compared to the past 1000 years, with drying trends generally in the west and wetting trends in the east. This study also found that seasonal shifts have occurred in some regions recently, with seasonality changes most notable for southern gauges. We expect that our new approach provides a foundation for incorporating various datasets to examine non-stationary variability in long-term precipitation climatology and to confirm the spatial patterns noted here in greater detail.

## 1 Introduction

Understanding meteorological drought trends is important as the entangled impacts of anthropogenic climate change and natural climate variability have complicated patterns of precipitation change over the last century (Ault, 2020; Schubert et al., 2016). Drought severity and duration has changed over time at seasonal, interannual or centennial scales, with subsequent impacts on human and ecological systems (Trenberth, 2011; Van Loon et al., 2016). Many studies have investigated trends or shifts in drought related to climate change (Marvel et al., 2021; Williams et al., 2020; Mishra et al., 2010; Marvel et al., 2019; Trenberth et al., 2014). Previous research has relied heavily on observed or remotely sensed precipitation records, which often

do not exceed 100 years. Although such observations can capture modern drought trends, 100 years of data are not sufficient for determining whether recent drought trends are a part of long-term cyclic variability, due to recent unprecedented trends, or a combination of the two (Easterling et al., 2000; Cook et al., 2015).

    In addition, previous studies have indicated that precipitation seasonality has changed during the observed period: those changes include increases in the amplitude between the wet and dry seasons, or temporal shifts in the driest/wettest period

(Marvel et al., 2021; Weiss et al., 2009; Pal et al., 2013). Even without substantial changes in the annual mean precipitation, shifts in precipitation seasonality can have significant impact on local ecosystems or man-made water management schemes like reservoirs that rely on storing and releasing seasonal flow. As a result, understanding seasonal cycles and non-stationary shifts in seasonality are important for building adaptive and robust water management schemes (Konapala et al., 2020). For climate projections of the next 100 years, Marvel et al. (2021) found projected changes in annual precipitation cycles across

the U.S. Midwest and Upper Great Plains. This region is projected to undergo a shift in peak precipitation to earlier in the year without substantial changes in precipitation. This study also projected an increase in precipitation during the wettest season (winter) in the Northwest and Southeast US, thereby increasing the seasonal variance in precipitation. Changes in seasonality or seasonal variance can be better understood when viewed in historical context using a much longer time window to determine whether they are within the range of natural climate variability or outside this range (Coats et al., 2015).

Therefore, much longer time scales are needed for a comprehensive understanding of non-stationary drought trends, preferably using a multi-centennial time scale (Torbenson and Stahle, 2018; Herweijer et al., 2007; Cook et al., 2010a; Diffenbaugh et al., 2015). Paleoclimate reconstructions use environmental proxies, such as tree-ring chronologies or speleothem records that physically record some aspect of climate, and can cover a much longer period than the instrumental observations (Cook et al., 2016). For example, this study uses a reconstruction of precipitation across North America based on tree-rings, which infer the

relative availability of regional precipitation or soil moisture from the annual growth. This particular reconstruction is a gridded continental-scale reconstruction, rather than a regional or local reconstruction. Large-scale gridded reconstructions sacrifice some local precision, but have the benefit of generating a single, complete dataset based on a common methodology, which can leverage a larger catalog of chronologies. Several such gridded hydrometeorological reconstruction datasets using tree-ring proxies are available across North America. The North American Drought Atlas (NADA; Cook et al., 1999) reconstructs

Palmer Drought Severity Index (PDSI; Palmer, 1965) in June to August (JJA) from 0-2006 A.D. and has been used to determine historic drought severities (Cook et al., 2010b; Cook and Krusic, 2008). The North American Seasonal Precipitation Atlas (NASPA) is another precipitation reconstruction recently developed with two distinct seasons: December to April (DJFMA) and May to July (MJJ) (Stahle et al., 2020). The NASPA dataset provides both SPI and averaged precipitation for both the cool and warm seasons. The NASPA is used here since it covers the past 2,000 years and contains cool and warm season

records for each year. Over 2,000 years of sub-annual scale records enable an investigation of non-stationary drought trends and seasonal shifts across multi-centennial scale if they can be combined with recent observed instrumental datasets (Trenberth et al., 2014; Marvel et al., 2019; Cook et al., 2016).

Despite the value of long reconstructions, comparing meteorological drought trends across observed and proxy-based reconstruction datasets is challenging as these data types are often not directly compatible (Baek et al., 2017; St. George et al., 2010). The first challenge is that each dataset often has non-negligible biases. Biases in proxy reconstructions can be caused by indirect measurement of the target variable, e.g., precipitation, by way of tree-ring growth. For example, bias can be introduced during the standardization process, designed to isolate the interannual signal from the long-term geometric growth of a tree. Trees also have physiological responses to continuous extreme drought or pluvials, which can limit variance at the extremes (Franke et al., 2013; Robeson et al., 2020). Even among gridded datasets based on gauge observations, bias can be introduced by the use of imperfect transforming algorithms (Sun et al., 2018), due to orographic induced bias, underestimation of trace precipitation amounts (Goodison et al. 1998), or wind-related undercatch (Pollock et al., 2018). Thereby, precipitation measurements for the same period can differ across datasets. These biases can cause one dataset to systematically under- or overestimate precipitation compared to other datasets (Robeson et al., 2020), or to modify the range of estimates. Quantifying and minimizing those biases is necessary to merge disparate datasets and analyze a common trend across various datasets.

A second challenge for merging reconstructions and observations is their heterogeneous spatial and temporal scales (Cook and Krusic, 2008). For example, the NASPA reconstructions provide only two time series per year with different precipitation periods: May-July and December – April. Instrumental datasets can have sub-daily, daily, or monthly temporal scales (Howard et al., 2021). Therefore, time scale must be unified if one is to merge instrumental and reconstructed datasets to observe common non-stationary seasonal trends. In addition, the spatial resolution of gridded datasets varies, and centers of those grid cells are not always matched. Thus, matching co-located grid cells through creating a common spatial resolution is an important aspect in representing common characteristics in precipitation (Abatzoglou, 2013).

This study is designed to address the challenge of constructing 2000 years of precipitation climatology by merging multiple datasets with varied biases and temporal scales. The objectives of this study are therefore to (1) construct downscaled NASPA precipitation time series from bi-annual into monthly scale with 3 months average resolution, (2) identify unique biases inherent in different precipitation data and remove those biases, and ultimately (3) construct a 2000-year continuous climatology model that can capture century scale shifts in the 3-month precipitation. This approach mimics the underlying distribution methodology of the Standard Precipitation Index. The continuous climatology derived from proxy reconstructions and modern observations is the true goal, with the first two objectives functioning as necessary intermediate steps towards this ultimate goal.

Here, we first temporally downscale NASPA data using a statistical downscaling technique, K-nearest neighbors (KNN). Then, we develop a model to simultaneously capture non-linear trends while accounting for unique biases across proxy and instrumental datasets by decomposing information from all datasets into their shared long-term trends, seasonality, and data-specific bias. Ultimately, our approach allows us to simultaneously model long-term trends in different seasons.

95

## 2 Methodology

For the first step, the temporal downscaling of NASPA precipitation, we applied the statistical downscaling technique, K-nearest neighbors (KNN). KNN is a statistical downscaling technique widely used in hydrologic time series (Raje and Mujumdar, 2011; Gangopadhyay et al., 2005; Gutmann et al., 2012) such as reconstructing annual streamflow from tree-ring chronology data or producing local-scale precipitation or temperature time series using neighboring climate stations (Gangopadhyay et al., 2005, 2009). A hierarchical Generalized Additive Model (GAM) model is then developed and applied to merge the datasets and analyze trends. This approach is tested at 14 sites across the continental US. Section 2.1 presents precipitation datasets used in this study, while Section 2.2 provides background on SPI calculation. Section 2.3 introduces the K-nearest neighbor (KNN) novel approach for temporal downscaling of the reconstructed precipitation and Section 2.4 describes the GAM model for merging disparate datasets and analyzing meteorological drought trends using the SPI framework.

### 2.1. Data

The Global Precipitation Climatology Center (GPCC; Becker et al., 2013) was used to temporally downscale/disaggregate the NASPA into monthly values as this was the instrumental target for the NASPA reconstructions. The underlying precipitation datasets used in the analyses presented here are as follows:

NASPA: The North American Seasonal Precipitation Atlas is a dataset of gridded reconstructions of precipitation, based on a network of 986 tree-ring chronologies from across the North American continent (Stahle et al., 2020). Precipitation totals and SPI are reconstructed for December–April (DJFMA) and May-July (MJJ) across a 0.5° x 0.5° grid, resulting in a total of 6812 grid cells (Stahle et al., 2020). The length of the reconstructions varies across space and between seasons but have a maximum of over 2,000 years at many locations, particularly in the western US.

The NASPA reconstructions target GPCC, applied at each GPCC grid point using ensembles of tree-ring chronology-based regressions (Stahle et al., 2020). An additional NASPA reconstruction dataset for MJJ exists for the period 1400-2016, in which the MJJ precipitation estimates were re-processed to remove any persistent signal from the DJFMA reconstruction (Torbenson et al., 2021). Our model uses the DJFMA and original, non-processed MJJ reconstructions to maximize the period of study and because the GAM model accounts for some level of persistence.

CRU TS: Climate Research Unit TS4.01 (CRU) is a 0.5° x 0.5° gridded dataset of monthly climate. It is based on individual station observations which are directly interpolated to a gridded scale (New et al., 2000; Harris et al., 2020). This study used version 4.01 which covers the period 1901-2018 (Harris et al., 2020). The CRU dataset was used because it is a well validated dataset that provides a long temporal coverage based on ground stations.

GridMET: The Gridded Observed Meteorological data (GridMET) is a gridded (1/24° x 1/24°) dataset of daily resolution available from 1950–2020, for the US (Maurer et al., 2002). GridMET is constructed by combining direct daily gauge observations with regional scale reanalysis to fill gaps (Abatzoglou, 2013). In this study, we assume the GridMET as a "ground

truth" and use it to correct biases in CRU and NASPA because the GridMET incorporates satellite data, making it highly accurate and spatially well-distributed with high resolution.

GPCC v7: The Global Precipitation Climatology Project (GPCC) is a gridded precipitation product built on gauge-based precipitation. The monthly-resolved GPCC v7 covers the period 1901 to 2013 at a 0.5° x 0.5° spatial resolution (Becker et al., 2013). Since the NASPA reconstructions were originally developed at a gridded scale via regression using GPCC data, and further validated and calibrated based on GPCC data, we assumed that the GPCC and NASPA datasets shares regional and temporal characteristics. Thus, this study uses monthly GPCC data to best mimic the intra-annual characteristics for temporally downscaling and disaggregating NASPA estimates to monthly time series. GPCC is only used for temporal disaggregation of NASPA data and is not included in the hierarchical GAM model.

## 2.2 Drought Measurement

Drought is defined as a lack of water within the hydrologic cycle relative to the given climatology of a location. Meteorological drought refers to a deficit of precipitation relative to typical conditions for a location and period. The severity of meteorological drought is often measured by the Standard Precipitation Index (SPI). The SPI is calculated by fitting n-days accumulated precipitation time series to a set of probability distributions for each period's climatology and then using these distributions to convert accumulated precipitation into the standard normal distribution (Lloyd-Hughes and Saunders, 2002; Stagge et al., 2017, 2015; Guttman, 1999). SPI values therefore represent the number of standard deviations from typical conditions for a site and time of the year. The SPI is widely used for studying or monitoring meteorological drought, particularly by the U.S Drought Monitor and World Meteorological Organization (WMO). It has unique strengths of using precipitation only: a simple data requirement and calculation process, and straightforward interpretation between averaged precipitation and drought severity (Dai, 2011; Ukkola et al., 2020; Svoboda et al., 2002).

In this study, we use a 3-month moving average of precipitation (SPI-3) to provide seasonal characteristics of drought (Patel et al., 2007). We present SPI-3 values of -1.5, 0, and 1.5, which are equivalent to the 6th percentile, mean, and 94th percentile thresholds of a fitted two-parameter Gamma distribution. These thresholds are commonly used in drought and pluvial studies to represent precipitation associated with dry anomaly, typical, and wet conditions (Heim, 2002), and a similar thresholds (5[th] percentile) is used by the US Drought Monitor (Svoboda et al. 2002).

## 2.3 Temporal downscaling using K nearest neighbor resampling

KNN is a downscaling technique designed to estimate some target information by searching a set of historical catalogs of the target vector and finding the k most similar analogs, where k can be any number of the user's choice (Gangopadhyay et al., 2005). In this study, monthly GPCC time series were used as sampling catalogs for selecting target vectors (annual precipitation sequences) based on NASPA values. More specifically, the goal is to insert K historical 13-month precipitation sequences from the GPCC library into a given year of the NASPA reconstruction based on similarity to the recorded SPI values during the prior and current year. The 13-month sequence is considered a single downscaling unit containing three known NASPA

values across a year (Figure 1). To do this, multiple (k = 10) annual historical sequences are inserted for each year of the reconstruction to approximate plausible monthly precipitation patterns that most closely match the three NASPA reconstructed periods.

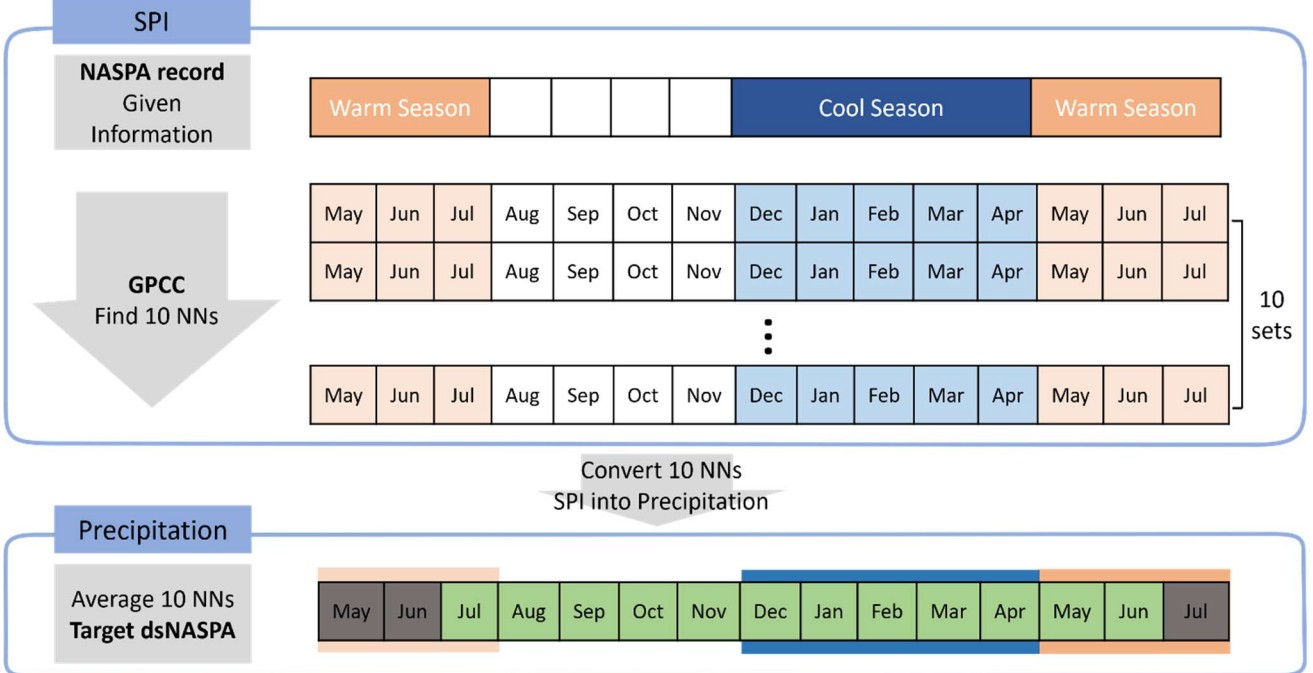


**Figure 1. A framework for the temporal downscaling process. Monthly scale NASPA (3-months averaged precipitation) time series is constructed using this process. This method is applied for every year of reconstruction.**

Figure 1 outlines the temporal downscaling process using KNN. For each year, NASPA values were constructed as the target vector using three data points: SPI-3 during the previous year's MJJ period, SPI-5 in this year's DJFMA period, and SPI-3 in this year's MJJ period. SPI-3 and SPI-5 values calculated from the GPCC instrumental period (1901-2013) for the same location were constructed as the data library. GPCC was used because it formed the basis for the original NASPA reconstruction (Stahle et al., 2020). Second, for each year, we calculated the Euclidean distances between the target vector from NASPA and the available GPCC library to select 10 sequences (k=10) from the GPCC SPI time series which have the closest Euclidean distance to the target NASPA SPI values. Note that resampled sequences are permitted to be any historical 13-months SPI series, regardless of whether the months align, increasing the number of available sequences from 113 (years in the GPCC dataset) to 1356 (years × months). This is possible because SPI is agnostic to season, each month follows a standard normal distribution. Then, the ten monthly resampled SPI-3 time series were converted back to the 3-months precipitation using 2-parameter Gamma distributions derived from the GPCC dataset. Lastly, the 10 sets of precipitation timeseries were averaged and inserted into the targeted year of the NASPA.

Overall, our downscaling approach provides a few advantages: first, it reflects the compatibility of the climate field as it searches analogs from the same location. Second, direct resampling based on similarity from the GPCC sample field incorporates realistic seasonal progression and the 3-month structural persistence of the SPI. Third, the K neighbors create an ensemble of equally likely time series, identifying an envelope of feasible time series when there is no information between the 3 points from the NASPA reconstruction.

Downscaling skill was measured by normalized mean absolute error (nMAE) using the following equations:

$$nMAE = \frac{\sum_{Month,year} |GPCC - NASPA|}{\sum_{Month,year} GPCC} \tag{1}$$

where GPCC represents the observed precipitation during the instrumental period and NASPA represents the ensemble mean of the reconstructed precipitation after applying the KNN downscaling to the NASPA reconstruction.

## 2.4 Bias correction using Hierarchical GAM

Generalized additive models (GAM) are statistical models that permit regression using non-linear smooth functions instead
of, or in addition to, linear covariates. GAMs are a subset of Generalized Linear Models (GLM), meaning their regression terms can represent parameters for data with distributions other than Normal. However, where most GLMs apply linear regression principles to model a distribution's parameters, GAMs can include non-linear terms (Simpson, 2018; Wood, 2008; Pedersen et al., 2019). When non-linear terms are applied to time series data, GAMs also permit spanning irregularly sampled data to model complex and non-linear drought trends. This method was applied to create a single, common estimate of the
temporally varying Gamma distribution parameters representing precipitation climatology by incorporating information from multiple biased data products. We refer to the process of accounting for seasonal bias in the mean and shape parameters from different data sets as "bias correction" for the remainder of this paper because it mirrors the process of bias correction by moment matching. However, unlike a separate bias correction step, this is performed within the GAM model, permitting confidence intervals around each of the bias correction terms.

GAM models have been previously applied to accumulated precipitation data to estimate the parameters of the 2-parameter gamma distribution SPI under non-stationary climate conditions (Stagge and Sung, 2022; Shiau, 2020; Sung and Stagge, 2022). This study relies on the non-stationary SPI approach introduced in Stagge and Sung (2022) and applied in Sung and Stagge (2022). In this approach, the two parameters (mean and shape) of Gamma probability distribution are modeled as slowly change through two covariates of time: year (to capture multi-decadal trends in certain month) and month (to capture recurring
seasonality). Here, we expand this approach, by adding a hierarchical grouping variable to simultaneously model common seasonal-specific long-term trends across datasets, while incorporating variability at the group level following the approach of Pederson et al. (2019). When applied, this model decomposes information from all datasets (CRU, GridMET, and NASPA) into a smoothed long-term trend that is common to all datasets and also an additional annual seasonality smoother that varies

slightly by dataset to account for bias relative to GridMET. In this way, there is a single common trend, with an adjustment
added to shift the mean and shape parameters up or down seasonally based on the data source.

The detailed model framework is shown below in Equations 1-3. The Gamma distribution is typically prescribed by shape and scale parameters ($\alpha$ and $\theta$, respectively), but our approach follows Wood (2006), instead estimating the mean and shape parameters (Eqs. 2-3). The scale parameter can then be estimated from the mean and shape (Eq. 1).

$$P_{3\ month,m,y} = gamma(\alpha,\ \theta)\ \begin{pmatrix} m: month\ of\ the\ year, \\ y: year \end{pmatrix} \quad where\ \ \theta = \frac{\mu}{\alpha} \tag{1}$$

$$\mu = \beta_{0\mu}\begin{pmatrix} CRU \\ NASPA \\ GridMET \end{pmatrix} + \beta_{1\mu}f_{s\_\mu}\left(X_{month}, by = \begin{pmatrix} CRU \\ NASPA \\ GridMET \end{pmatrix}\right) + \beta_{2\mu}f_{te\_\mu}(X_{year}, X_{month}) \tag{2}$$

$$\frac{1}{\log(\alpha)} = \beta_{0\alpha}\begin{pmatrix} CRU \\ NASPA \\ GridMET \end{pmatrix} + \beta_{1\alpha}\ f_{s\_\alpha}\left(X_{month}, by = \begin{pmatrix} CRU \\ NASPA \\ GridMET \end{pmatrix}\right) + \beta_{2\alpha}\ f_{te\_\alpha}(X_{year}, X_{month}) \tag{3}$$

where $P_{3\ month,m,y}$ represents the 3-month moving average precipitation at year $y$ and month $m$. The precipitation is fitted in Gamma probability distribution, where have $\mu$ (mean) and $\alpha$ (shape parameter). The $\beta$s are different parameters of each spline function, $f_s$ and $f_{te}$, which denote cyclic and tensor spline, respectively. The underlying principle of the model is that there exists a single best estimate of the precipitation distribution at any given time, described by the mean and shape parameters of the gamma distribution that changes seasonally $\beta_1 f_s(X_{month}, by = dataset)$ and can also change slowly on a multi-decadal
scale, $f_{te}(X_{year}, X_{month})$ with constant y intercept, $\beta_0(dataset)$. Similar to quantile mapping bias correction (Lanzante et al., 2018; Ho et al., 2012), the $\beta_{0,model} + \beta_1 f(X_{mont}, by = dataset)$ terms in both shape and mean parameters allow for adjustments based on the month and the model. The model is therefore capable of modeling trends and correcting data-induced bias simultaneously.

The single common tensor product spline smoother ($f_{te}(X_{year}, X_{month})$) is shared across all datasets to model the interaction of
long-term trends ($X_{year}$) relative to season ($X_{month}$) using smoothly changing parameters for the two dimensions (year and month). A tensor product spline is an anisotropic multi-dimensional smoother, meaning it can model the interaction of variables with different units and can assign different degrees of smoothing for each direction, as is necessary for dimensions of month and year. Estimating $\beta_{2\mu}$ and $\beta_{2\alpha}$ in terms of year and month allows for non-linear annual trends for each month while constraining these trends to be smooth through time. We constrain the smoother with control points (knots) every 70 years for mean and
shape parameters to approximate climate variability on decadal scales while preventing excessive sensitivity/volatility. As such, the tensor product can simultaneously model seasonal precipitation regime, shifts of those periods to earlier or later in

the year, and non-stationary changes in the long-term. The tensor spline approach to model trends in two time dimensions follows the methodology of Stagge and Sung (2022).

The first two terms derive intercept and seasonality distinctive to each dataset. The first term ($\beta_0\ (datasets)$) accounts for dataset specific intercept and the second term ($\beta_1 f_s(X_{month}, \text{by} = dataset)$) accounts for dataset specific seasonality. Cyclic spline functions ($f_s$) were applied for modeling the seasonality term assuming a cyclic function for the recurrent monthly term constrains the model so that late December and early January have similar values, matching up to their second derivative. This term is stationary, i.e., does not change year to year. The *f(X_{month} by=dataset)* term uses group level smoothers for this seasonal bias spline, so that each dataset applies unique seasonal adjustments to the common tensor product spline. A dataset-specific intercept, *β₀*(dataset) was also included to capture consistent biases between datasets. The variations of smoothing functions and parameter *βₛ* are modeled using 'mgcv' packages in R (Wood, 2008).

Bias correction was conducted based on three assumptions: (1) the GridMET dataset is not systematically biased (Yang et al., 2017), (2) the magnitude of bias can differ by season, and (3) biases are stationary in the long-term, i.e. biases during overlapped periods are representative of biases throughout the rest of the data. Following the first assumption, when plotting results, we adjust CRU and NASPA parameters to match the GridMET dataset. The second and third assumptions are addressed by the $\beta_{0,model} + \beta_1 f(X_{mon}, by = dataset)$ permitting different bias corrections by month and model, which are estimated during overlapping periods and fixed outside this period.

The significance of the modeled trend is tested using the instantaneous first derivative method. This method calculates the first derivative of modeled trend with 1000 randomly drawn estimates of the modeled mean and shape parameters through time (by year). Then, we calculate 95% confidence interval around the first derivatives to indicate periods where the trend is significantly different from zero, i.e., the trend is increasing or decreasing. The non-linear trend analysis approach overcomes the limitation of simple linear significant tests which only capture monotonic changes. In doing so, it is not possible to discuss a single "trend", but once can discuss whether the distribution mean is significantly increasing or decreasing at a given time, represented by the instantaneous first derivative. As such, this method has the benefit of preserving all non-linear and non-stationery characteristics in modeled trends, while providing estimates of significant changes. The results of this analysis are shown in Fig. S5.

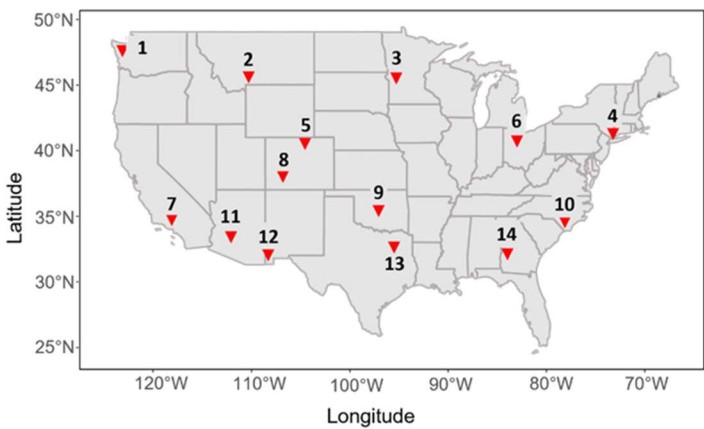

Figure 2. Gauge site locations. The abbreviation of each locations are as follows.1: Aber,WA. 2. Grd,MT. 3. Mor,MN. 4. Nyc,NY. 5. Den,CO. 6. Mrv, OH. 7. Los, CA. 8 Mtn, CO. 9. Okc,OK. 10.Sbw, NC. 11. Phx, AZ. 12. Roe, NM. 13. Wax, TX. 14. Alb, GA

The developed model was applied to 14 locations across the continental United States (Figure 2; Table 1). These sites were chosen based on relatively long instrumental records, adequate NASPA reconstruction skill, and to represent a wide range of climate regions. NASPA reconstruction skills are investigated via calibration and validation statistics by data creators (Stahle et al., 2020). One of the calibration statistics, the coefficients of multiple determination ($R^2$) are presented in table 1. We avoid determining whether the datasets are acceptable or not through these statistics, rather to clarify which seasons or regions have better skills.

Table 1. List of sites considered in this study. The number for each site refers to the location in Fig. 1. NASPA reconstruction skill for the cool (DJFMA) and warm (MJJ) seasons are presented as R2.

|  | Symbol | Location | NASPA Skill($R^2$) | |
|---|---|---|---|---|
|  |  |  | DJFMA | MJJ |
| 1 | Aber, WA | Aberdeen, Washington | 0.457 | 0.485 |
| 2 | Grd, MT | Gardiner, Montana | 0.486 | 0.485 |
| 3 | Mor, MN | Morrisville, Minnesota | 0.347 | 0.478 |
| 4 | Nyc, NY | New York City, New York | 0.27 | 0.437 |
| 5 | Den, CO | Denver, Colorado | 0.346 | 0.671 |
| 6 | Mrv, OH | Marysville, Ohio | 0.295 | 0.323 |
| 7 | Los, CA | Los Angeles, California | 0.782 | 0.533 |
| 8 | Mtv, CO | Monte Vista, Colorado | 0.446 | 0.498 |
| 9 | Okc, OK | Oklahoma City, Oklahoma | 0.441 | 0.428 |
| 10 | Sbw, NC | South Brunswick, NC | 0.245 | 0.265 |
| 11 | Phx, AZ | Phoenix, Arizona | 0.58 | 0.371 |
| 12 | Roe, NM | Rodeo, New Mexico | 0.544 | 0.366 |
| 13 | Wax, TX | Waxahachie, Texas | 0.365 | 0.644 |
| 14 | Alb, GA | Albany, Georgia | 0.508 | 0.220 |

# 3 Results

## 3.1 Temporally downscaled monthly NASPA time series

In order to merge the NASPA data with CRU and GridMET, the irregularly spaced NASPA must first be temporally
downscaled, or disaggregated, to a regular monthly time step and 3-month duration. The downscaled NASPA (dsNASPA) 3-
months averaged time series was constructed at a monthly scale and given the ds- prefix to distinguish it from the original
NASPA reconstruction. Figure 3 shows three example years for two sites with very different climatology, showing the
ensemble of 10 selected nearest neighbors (pink), the resultant dsNASPA estimate (black), and the true value from the GPCC
for those years when data is available (1950 and 2010, blue). Each figure displays 13 months, or one unit of the KNN
downscaling process, from the previous year's July to the current July. The downscaling results at all study sites are shown in
Figure S1 and S3.

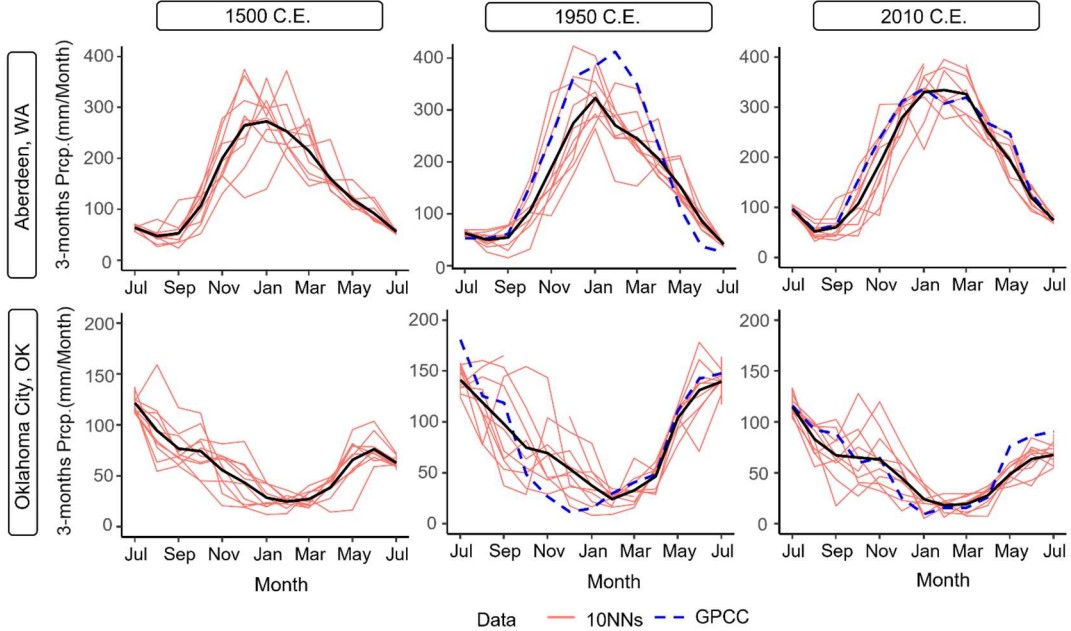

**Figure 3.Comparing downscaled NASPA (Black), 10 Nearest Neighbors (pink) and GPCC (blue dashed) at Aberdeen, WA (upper)
and Oklahoma City, OK (lower). The marked month in X-axis refers the last month of 3months-rolling average.**

The dsNASPA generally agrees with the GPCC, especially in capturing seasonality (Fig. 3 and Fig. S1). Downscaling skill is
generally good in the season between DJF-MJJ where the NASPA reconstruction covers all three months (Fig. 3 and Fig.4).
July SPI-3 (MJJ) often produces the smallest nMAE, which is logical given that the July SPI-3 period exactly overlaps with
the warm season MJJ from the NASPA Thus, the downscaling process has good information during this period and is not
required to do as much.

There are a few exceptions showing the best skill during winter (DJF) and poor skill, large nMAE, during early summer (MJJ, Figure 4). This occurs only in the southwestern US (e.g. Los, CA, Phx, AZ and Roe, NM) where the underlying NASPA shows better initial reconstruction skill during their relatively mild winters (Table 1) and less skill in summer. For the dsNASPA, the seemingly large errors in nMAE during MJJ are primarily due to extremely small values in the denominator of the nMAE due to very low precipitation, combined with not capturing infrequent large rainfall events. The Figures in S3 illustrate a few large precipitations in these regions drive large nMAE (scatter plot), however, dsNASPA still well matches with GPCC (time series). Despite the limitation, downscaling accurately predicts the general precipitation pattern in terms of seasonal and long-term average precipitation, with nMAE values generally between 0.1 - 0.5. We compared performance of the dsNASPA with a highly naive alternative (assuming the mean of GPCC climatology) and found that dsNASPA provides a clear signal in the period with NASPA information (blue shaded period in Fig 4). As expected, the dsNASPA provides less information in the interpolation period where NASPA estimates are not available. However, during the gap seasons, the dsNASPA still produces positive correlation with observations, useful for measuring climatological shifts, and greatly reduces extreme errors created by the naive estimator in the semi-arid West. For regions other than the semi-arid locations described above, errors during periods of good NASPA coverage occur primarily due to the errors between sampled GPCC and NASPA (Figure S3). For example, the dry bias shown in July (MJJ) between dsNASPA and GPCC in 1950 at Oklahoma City (Figure 3, center) is caused by uncertainty in the original NASPA dataset, which caused the converged point of nearest neighbors (black) to underestimate precipitation relative to the GPCC observation (blue). May and June (MAM and AMJ, respectively) are reproduced nearly as well, given that these periods share coverage from the cool (DJFMA) and warm (MJJ) reconstructions. Later periods of the 5-month cool season reconstruction, March (JFM) and April (FMA), show reasonably good accuracy. Error increases through fall and winter across a temporal gap between NASPA reconstructions (Fig. 4 and Fig. S3). This is indicated by much broader resampled estimate ranges (Fig. 3 and Fig. S3) during the late fall and early winter.

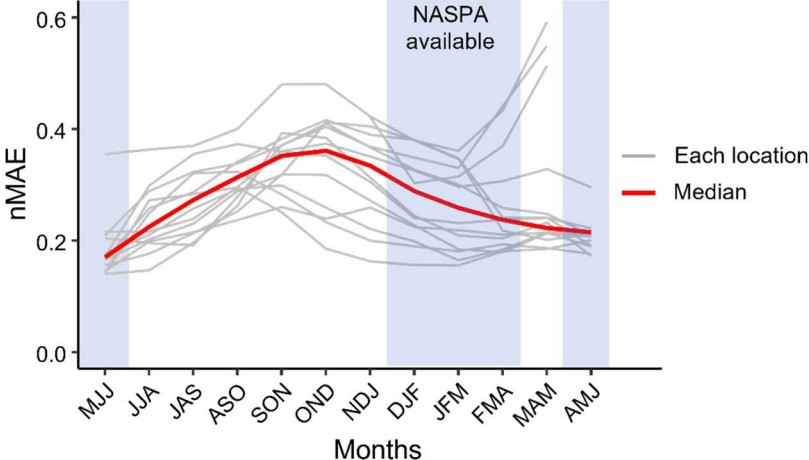

**Figure 4. nMAE to indicate downscaling skills at each location and its median.**

**3.2 Investigating model bias**

Here we investigate how dataset bias is quantified in the model. As mentioned, our model accounts for two types of bias: a consistent bias for a given dataset across the entire year and seasonal specific bias. These bias terms were estimated for both the mean and shape parameters.

Typical results for the Monte Vista, CO gauge show how these biases are captured in a single model. Fig. 5 shows the non-stationary mean estimate for each dataset, represented by colored lines, and the range from SPI = -1.5 to +1.5 as grey shaded

regions. Note that the non-stationary mean lines all follow the same trend, simply adjusted up or down based on bias. At this station, dsNASPA and CRU tend to underestimate precipitation relative to the GridMET benchmark across all four seasons. This consistent offset may be due to the significantly coarser resolution of NASPA and CRU, which may not capture elevation effects, particularly in this mountainous region. The magnitude of bias also differed by season in this example, with the greatest differences visible during the ASO season (Fig. 5). Note that the four periods we highlight in this study were purposefully

chosen to mimic NASPA availability, anchored by the MJJ 3-month period, rather than the more commonly used seasons (DJF, MAM, JJA, SON).

In addition to bias in the mean parameter, it is possible to detect model bias in the shape parameter, which controls variance, and thus the range between SPI = -1.5 and +1.5. The most notable bias in the shape parameter for the Monte Vista, CO example is for the dsNASPA, particularly during ASO, where the shape parameter is significantly overestimated, thereby decreasing

the variance for the same mean (Fig. 5). This is logical, as the ensemble resampling approach likely decreased extremes for the ASO period where there is no direct NASPA information. The shape parameter bias is negligible for the FMA and MJJ periods, which have full NASPA coverage. Shape parameter bias results for Monte Vista, CO is typical of other gauges studied here, with largest bias during the interpolated ASO period and little bias in periods with good NASPA information.

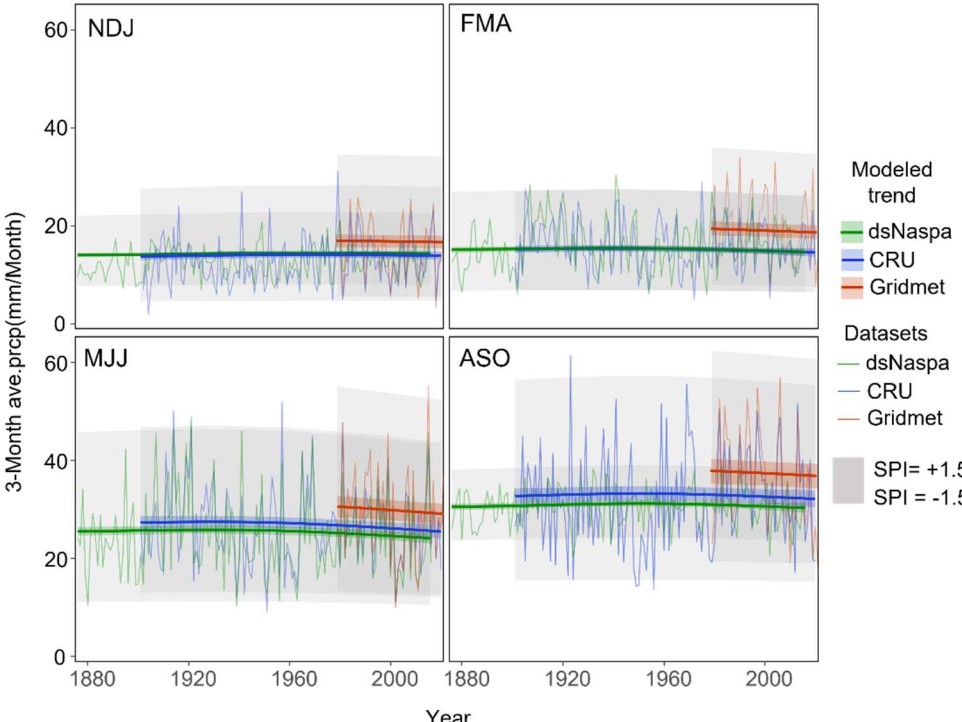

**Figure 5. Long-term trends of each dataset for 4 periods at Monte Vista, CO. The modeled long-term trends from each dataset are represented as differently colored lines, while the SPI -1.5 to 1.5 range for each dataset is indicated by grey regions.**

We present the results for all other regions in Fig. S4. The results indicate that the shape biases are largely dependent on the season, whereas mean biases are more dependent on the gauge. Notably, the ASO season shows large biases in the shape parameter. This is primarily because the dsNASPA in this season can't represent occasional extreme precipitation values, inducing an underestimation of its variance. In contrast, the MJJ season shows considerably less bias since the dsNASPA was developed from complete precipitation estimate in NASPA. A few exceptions exist in Mtv, CO and Grd, MT, where have large biases in the mean parameters across all seasons, possibly due to topographic effects between the gauge locations in these mountainous regions.

### 3.3  Constructing long-term trends

By accounting for the model-induced bias described in Section 3.2 and adjusting all datasets to match GridMET, we were able to generate a 2000-year model of non-stationary precipitation trends for each gauge. The modeled long-term trends incorporating bias correction across all instrumental and proxy datasets in Albany, GA and Monte Vista, CO are presented in Figure 6 as examples to illustrate the results of this approach. Figure 6 represents the long-term mean for each season as a line with a shaded range between SPI of -1.5 and +1.5, similar to Figure 5. The solid black line shows the common, long-term

trend of the mean while the precipitation series are shown as raw data without bias correction for context. Fig. 6 focuses on the period 1400 – 2020 year when the original NASPA dataset has the best reconstruction skills (Stahle et al., 2020).

    It is noteworthy that all seasons in Albany, GA have experienced noticeable trend changes in recent years, but the direction of change differs by season. Figure 6a shows the warm season (MJJ) has undergone a long-lasting wetting trend from the 1800s to 1900, followed by a drying trend during the 20th century in both the mean (SPI = 0) and wet anomalies (SPI = 1.5). NDJ

shows a wetting trend beginning in the mid-1800s and continuing to the present for both wet and dry anomalies. The NDJ mean in current years (2000-2020) is the wettest condition of the last 1000 years (Figure 6a). This agrees with previous findings using the NASPA dataset which have identified the southeast US including Albany, GA as experiencing the greatest positive precipitation trend during the DJFMA cool season (Stahle et al., 2020).

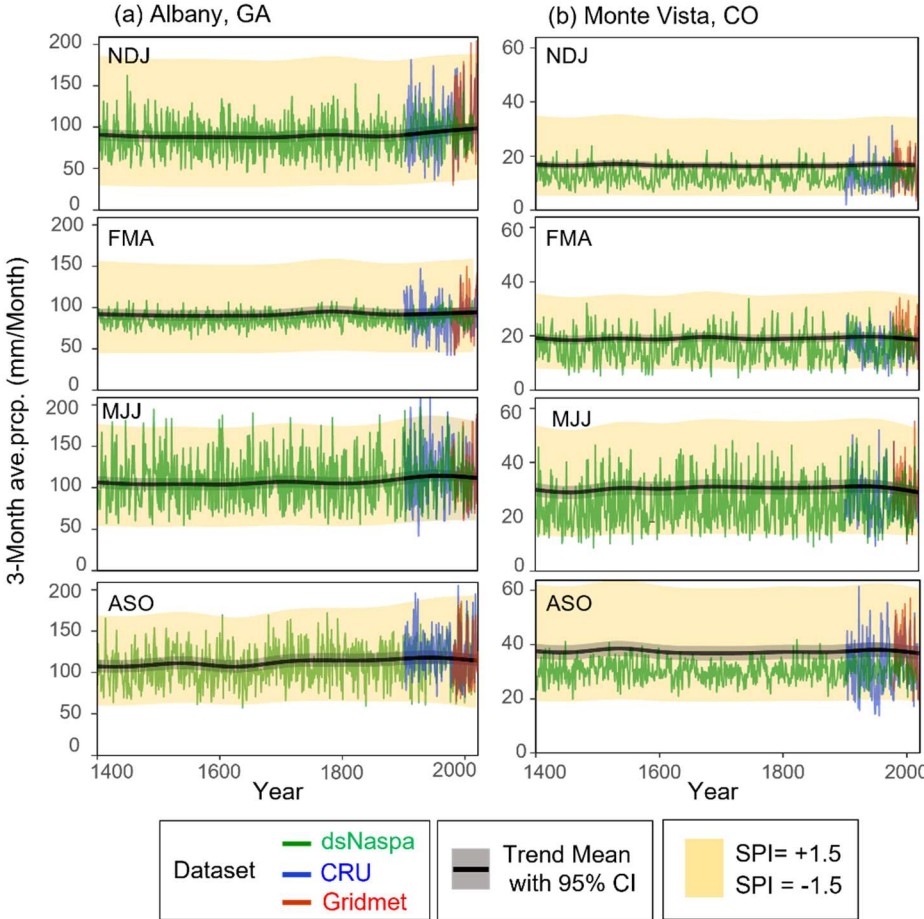

**Figure 6. Long-term trends in 3 months NSPI (black line) and averaged precipitations in four periods. The yellow shaded area represents the precipitation amount between SPI = +1.5 (upper boundary) and -1.5 (lower boundary) in the fitted Gamma distribution. Trends in the mean and SPI = -1.5 to +1.5 range are shown using GridMET as baseline to illustrate bias correction, while the raw data is shown without bias correction for context.**

We note that the Albany, GA, site has also experienced changes in the magnitude of variability between dry and wet extremes.

The range between SPI = 1.5 and SPI = -1.5 became much larger during the recent period, particularly for the ASO season, implying both wet and dry anomalies have become more extreme than during prior centuries. The strong drying trend in MJJ coupled with a wetting trend during the NDJ season indicates a seasonal shift of the driest season. While NDJ has historically been the driest period among the 4 seasons, during the modern period, MJJ now has similar dry conditions to the NDJ period. Monte Vista, CO had very stable SPI trends until the 19th century before undergoing a rapid drying trend during the 20th

century, particularly during the MJJ and ASO periods (Fig. 6b). The modeled MJJ precipitation at normal climatology (SPI = 0) is currently at its driest value in approximately 500 years after a long stable period between 1500 and 1900. A red line is shown as a baseline in Fig. 5b to indicate the modern MJJ mean for comparison against the reconstructed mean. ASO also shows drying trends within the last 300 years.

We can see that the modeled mean of dsNASPA in Monte Vista, CO is shifted upwards from its original average value because

of bias correction adjustment to match slightly drier dsNASPA climatology with the slightly wetter GridMET climatology for this site (Fig. 5). Our modeling process detected those biases and calibrated to shift upwards while maintaining a common gradual trend.

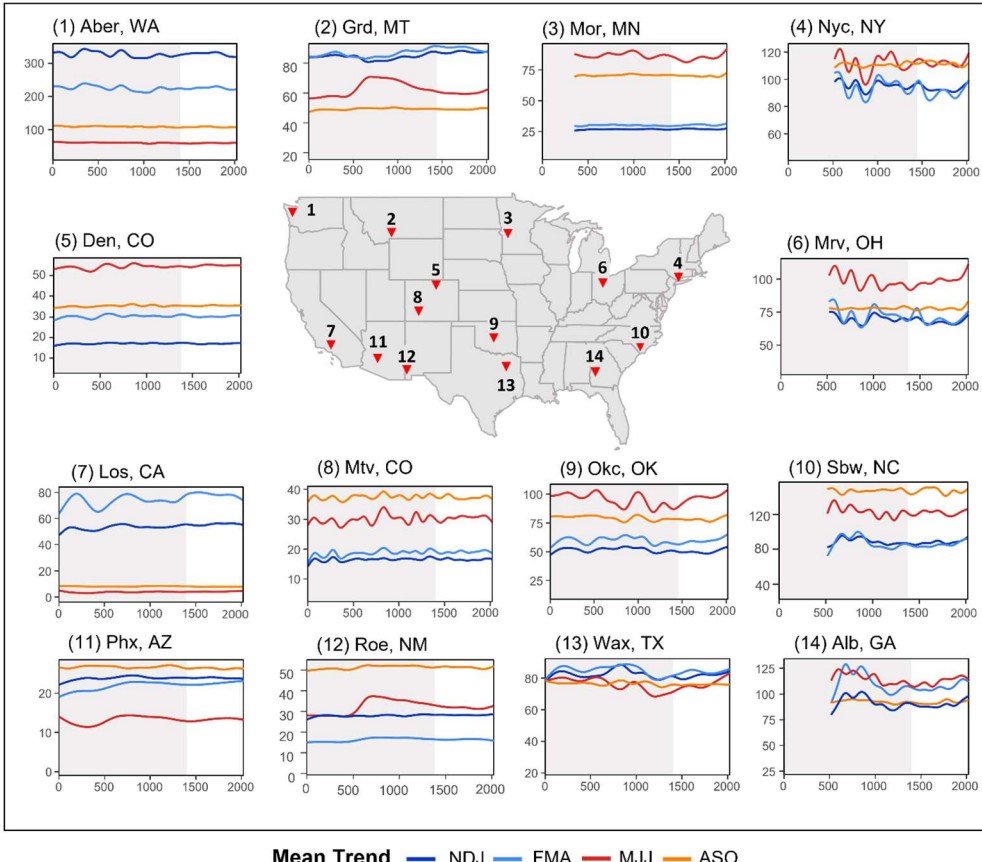

**Mean Trend** ── NDJ ── FMA ── MJJ ── ASO

**Figure 7. Long-term trends of averaged 3 months precipitation(mm/month) for 4 seasons. The periods before 1400 CE are shaded to represent the period with less prediction skills in original NASPA.**

Figure 7 provides long-term trends the four seasons previously discussed across all 14 study sites. The statistical significance of those changes is observed in Fig. S4. The plots show natural seasonality as differences between seasonal lines and long-term climate non-stationarity changes in each line. This separation allows for an evaluation of recent precipitation trends by comparing the past 100-year trend with the longer 2000 year time window. While results from the entire non-stationary GAM model are presented in Figure 7, extending back to the earliest NASPA reconstructions, our primary focus is on the period after 1400, shown in white. Prior to 1400 CE the NASPA reconstruction has greater uncertainty, and so is provided here for full context, but shaded in grey to emphasize this greater uncertainty. Figure 7 shows that the 14 demonstration sites generally follow a spatial climate pattern found in previous studies: with industrial era drying trends in the southwestern US, and wetting trends in eastern US (Lehner et al., 2018; Prein et al., 2016; Ellis and Marston, 2020). The drying trend in the west is most prevalent during each sites' wet seasons, with smaller or negligible trends during the driest part of the year. For example, the wet season drying trend is visible in Aber, WA, where after several centuries of stable precipitation there has been a decrease during the cool, wet seasons (NDJ and FMA). The wet season (FMA) in Grd, MT and Los, CA, also show clear drier trends during the most recent century or more. The drier trend in Los, CA during FMA precipitation has declined since 1500 CE but

this trend was exacerbated and became more severe during the 20th century, effectively shortening the winter wet period prior to the region's dry summer. The 20th century drying trend in Mtv, CO, has occurred across all seasons, not only for the wettest period like the other western stations. The most severe drying trend occurred in MJJ as mentioned in previous section (Fig. 6). The most severe western sites illustrate the value of comparing 20th century drying trends to longer reconstructed records to identify rapid and exceptional precipitation changes. Unlike these western sites, Den, CO, shows negligible long-term trends,

while the desert southwest (Phx, AZ and Roe, NM) exhibit minor wetting trends which are largely within the pre-industrial historical range.

The eastern part of the US generally has experienced rapid wetting trends during the most recent century as observed in previous research (Bishop et al., 2021). Those wetting trends are especially drastic in Nyc, NY, Mor, MN, and Mrv, OH, each currently experiencing the wettest conditions of the last 500 years of pre-industial, presumed near natural cyclic variability.

This pattern is particularly visible for the warm, wet summer season (MJJ). Sbw, NC also indicates a wetting trend in all seasons since 1700 CE, but those trends are not as rapid as the more northern sites. Warm season (MJJ) precipitation has also increased in the southern plains (Okc,OK and Wax,TX). Precipitation during the summer season has been gradually increasing since 1400 CE, but has undergone far more rapid increases since 1900 CE. Note that non-stationarity in the eastern US is less stable, which may be related to greater uncertainty in the NASPA reconstructions for this region with generally poorer

reconstruction skill (Table 1, Stahle et al., 2020).

Some locations have experienced different direction of changes based on seasons. Those changes mostly occur in southern part of the US as shown in Fig. 7 (11) – (14). For example, FMA precipitation in Phx, AZ has become slightly wetter during the last 500 years, while the other three seasons have been slightly drier during the 20th century. Alb, GA shows recent drying trends during the spring and early summer (FMA and MJJ) but wetter trends during fall and winter (ASO and NDJ). Those

seasonal specific changes ultimately shift the timing of the wettest or driest season. For example, while the NDJ season has been the driest season during the past 500 years, slightly drier than the preceding ASO season, this has changed during the 20th century. In addition, the difference between the wet seasons (Feb-Jun) and dry seasons (Aug-Jan) is decreasing as the wet seasons become drier and dry seasons become wetter. Seasonal shifts also appeared in Wax, TX, stemming from a constant wetting trend of MJJ season since 1000 CE while other seasons have experienced what is presumed to be the natural variability

with no abrupt 20th century changes, although further analysis is required to quantify the impact of natural variability. Though not the focus of this study, our results do capture past drought events such as the prolonged dry period in 1200 -1300s in Okc, OK and Wax, TX, consistent with previous studies regarding the so-called Medieval mega drought (Stahle, 2020; Cook et al., 2016).

## 4 Discussion

Our novel approach for temporal downscaling, combined bias correction, and non-linear trend modeling enables analyses of meteorological drought changes at a multi-centennial scale. Our downscaling approach allows irregular historical reconstruction to be included with instrumental records in a single long-term trend model using the same temporal scale, and

ultimately to compare non-stationary drought trends across seasons. The KNN downscaling approach preserves greater certainty during seasons with NASPA reconstructions and wider uncertainty during seasons that must be interpolated.

Simultaneous temporal trend fitting and bias correction, constrained with a GAM spline model, appears to provide a stable framework to merge these disparate datasets.

When developing the KNN approach, we chose to consider 13-month time segments regardless of seasonality, which may not capture some higher order characteristics like seasonal correlation. This design decision was a trade-off between the benefits of a larger sampling library of feasible SPI traces and the risk of overlooking some seasonally-specific time series behavior.

We chose the former, with an additional assumption that anchoring the time series behavior at three seasonal points would likely oversample segments with similar seasonal behavior. Also, our process of selecting SPI sequences and converting back to precipitation based on the seasonal probability distribution reflects the region's seasonal characteristics. This is demonstrated in Fig. S1 showing that our dsNASPA captures the general seasonality well. Still, future research might explore the magnitude of seasonality effects and persistence on SPI sequences in the downscaling process.

We also acknowledge the uncertainties over whole modelling process. The uncertainties in dsNASPA stemming from the downscaling process in addition to the original reconstruction process varies by region and season. As shown in our results, the downscaling skills are much higher in the period where the original NASPA provides information (Figs. S1 and S4). Reconstruction skills in the original NASPA vary depending on the region. This can be investigated in previous study and data source (Stahle et al., 2020).  In addition, stationarity in bias is an assumption of this method, however, it is a necessity that

underlies most proxy reconstructions. Based on prior NASPA validation, we are most confident that bias remains consistent during the period with consistent tree-ring coverage (1400-present, shaded while in Fig. 7), but may begin to change as chronology coverage and reconstruction skill decreases (beyond 1400, shaded grey in Fig. 7).

Nonetheless, our primary objective is a realization of the best possible estimate of the changes in precipitation distributions (climatology) of the past, rather than to replicate specific events in the time series. If one was interested in predicting

precipitation in a given year, we recommend using the original NASPA dataset, rather than our dsNASPA interpolation. Overall, our derived Gamma distribution can be used to understand the most likely climatology of the past, and potentially, the future based on available data.

In this context, our KNN approach creates a plausible estimate for periods lacking NASPA estimates (e.g., ASO) and is apt for estimating smooth changes in distribution of historical climatology. As further support, we found that taking the mean of

neighbors ensembles did not artificially and dramatically decrease the variance (Fig. S1).

In particular, this approach appears successful at generating a continuous sequence of bias-corrected precipitation distributions while addressing some of nuances of the NASPA reconstruction. The NASPA contains internal bias in reconstructing precipitation for the cool and warm seasons, with skill varying across the continent. Further, within the cool or warm period, the reconstruction can be dominated by one or two months. For example, the DJFMA reconstruction in California displays

significantly higher correlations with December, January, or February precipitation totals than for March or April. However, DJFMA was used as a compromise to ensure a common cool period across the US. It appears that hierarchical GAM bias

correction combined with KNN downscaling mitigates some of this effect by creating a local model for each site. Further, by using a seasonally varying bias correction, the model adjusts to the months, for example within the DJFMA, that are best captured.

We found a general drying trend for the wettest seasons in the western US and wet trends across most seasons in the eastern US. For some of these sites, 20th century trends appear to be rapid and outside the range of the long-term reconstructed record, whereas for other sites these patterns could be considered within the pre-industrial range and perhaps part of natural climate variability. Our results also pointed out some study sites where precipitation trends differ by season, leading to slightly altered seasonality.

Results for the case study at Los Angeles, CA agrees with general findings that showed extraordinary drying trends in the western US during the last century following a prolonged period of stable precipitation patterns since the 1500s (Stahle, 2020) The previously documented Medieval era megadroughts in the Great Plain region (Cook et al., 2016) also appear in our results. This consistency of results indicates that incorporating NASPA reconstructions data using our new method is feasible and can be useful to identify low frequency droughts trends and variability during the past 2000 years.

By contrasting the severity of precipitation changes during the past century with 2000 years of data, this model provides a potential to analyze the magnitude of recent trends during the modern increase in greenhouse gases with pre-Industrial natural variability. For example, Figs. 5, 6b and 7 each present subfigures showing the same meteorological drought trend model results at Monte Vista, CO. Each progressive figure takes a wider temporal viewpoint, from the last 120 years (Fig. 5), the last 600 years including the pre-Industrial period (Fig. 6b), the longest possible view beginning in 0 CE. As can be seen, the drying

trends in Figure 5 are rather steadily decreasing, but do not capture its extraordinary changes shown in Figure 6b. Taking a longer perspective implies that the modern 120-year data period is outside of the 'pre-industrial levels' defined by UN Paris agreement (IPCC, 2014), with the modern MJJ mean at its driest in 600 years. Our results agree with other findings that have identified recent or projected future shifts in seasonal precipitation (Marvel et al., 2021) or enhanced precipitation variability (Williams et al., 2020) due to anthropogenic climate change. For example, for several western sites, this study observed rapid

drying trends during the wettest seasons (Aber, Mtv, Grd, Los). These wet seasons are particularly critical in the western US, which relies on seasonal precipitation to fill reservoirs for later use during dry seasons. We therefore believe that modeling these smooth seasonal shifts over multiple centuries can inform water management plans to adapt to a changing climate. In addition, we believe understanding the seasonal specific changes in meteorological drought can help to analyze seasonal shifts in other types of droughts, such as anticipated summer soil moisture drought southwest US due to declining spring precipitation

(Williams et al., 2020).

We expect that our approach described here as a model validation for several case study sites could be applied across a denser network of sites to determine how meteorological drought has changed during the modern instrumental period and to put these trends into a much longer, pre-Industrial context. A unique benefit of this approach is that it models non-linear changes in typical precipitation (SPI=0), dry anomalies (SPI < 0), and wet anomalies (SPI > 0) simultaneously across all seasons.

## 5 Conclusion

This study introduced a novel method designed to apply the recently non-stationary SPI approach (Stagge and Sung, 2022) to a multi-century temporal scale by merging disparate datasets with a common tensor product spline term. To accomplish this objective, first we downscaled the irregularly spaced, bi-annual NASPA reconstruction into 3 months average precipitation with monthly resolution using a KNN approach. This permits analyses at a seasonal scale and enables the NASPA reconstruction to be integrated with instrumental data. In accordance with FAIR data principles, we make our data publicly available to allow researchers to access it and develop future drought trend studies (Wilkinson et al., 2016).

Second, we identified unique biases arising from different precipitation data sources and accounted for these biases in a hierarchical GAM model with model-based bias correction. This model corrected both persistent biases and seasonal specific biases in both mean and shape parameters of fitted distributions. Accounting for unique seasonal biases is important as previous studies have found bias magnitude can vary by season (Piani et al., 2010; Li et al., 2010). This is especially relevant when merging NASPA with observation datasets, because the temporal downscaling procedure depends strongly on season, e.g., MJJ is made directly from original reconstruction while ASO is based on KNN interpolating between the prior MJJ and the future DJFMA.

Third, after confirming that the temporal downscaling and non-stationary SPI model with bias correction were able to capture long-term trends, this study applied the model to a wide range of case study sites. Analyzing long-term trends in each season permits observation of shifts in seasonality and its variability. Those changes are also captured by season, so that our study could point out a specific season that is experiencing rapid changes although other seasons do not have drastic changes.

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
