# Peer review of "Assessing decadal to centennial scale nonstationary variability in meteorological drought trends"

_EGUsphere, 2022_

## Referee Comment (RC2)

[referee-annotated manuscript omitted]

---

## Author Comment (AC1)

*Supplement of*

**Assessing decadal to centennial scale nonstationary variability in meteorological drought trends**

Sung et al.,

*Correspondence to*: Kyungmin Sung (sung.229@osu.edu)

[Figure]

*Figure S 1. Compare the distribution of precipitation in GPCC VS 10 nearest neighbors.*

[Figure]

*Figure S 1 (Continue). Compare the distribution of precipitation in GPCC VS 10 nearest neighbors.*

[Figure]

*Figure S 2. nMAE (Normalized Mean Absolute Error) of dsNASPA compared to GPCC*

[Figure]

*Figure S 3. (upper) Compare the precipitation time series of GPCC (red). downscaled NASPA (black) and its nearest neighbors(sky), (lower) GPCC VS downscaled NASA*

[Figure]

*Figure S 4 (Continue). (upper) Compare the precipitation time series of GPCC (red). downscaled NASPA (black) and its nearest neighbors(sky), (lower) GPCC VS downscaled NASA*

[Figure]

*Figure S 3 (Continue). (upper) Compare the precipitation time series of GPCC (red). downscaled NASPA (black) and its nearest neighbors(sky), (lower) GPCC VS downscaled NASA*

[Figure]

*Figure S 3 (Continue). (upper) Compare the precipitation time series of GPCC (red). downscaled NASPA (black) and its nearest neighbors(sky), (lower) GPCC VS downscaled NASA*

[Figure]

*Figure S 3 (Continue). (upper) Compare the precipitation time series of GPCC (red). downscaled NASPA (black) and its nearest neighbors(sky), (lower) GPCC VS downscaled NASA*

[Figure]

*Figure S 3 (Continue). (upper) Compare the precipitation time series of GPCC (red). downscaled NASPA (black) and its nearest neighbors(sky), (lower) GPCC VS downscaled NASA*

[Figure]

*Figure S 3 (Continue). (upper) Compare the precipitation time series of GPCC (red). downscaled NASPA (black) and its nearest neighbors(sky), (lower) GPCC VS downscaled NASA*

[Figure]

*Figure S 4. Modeled mean, drought and pluvials anomaly (SPI = -1.5 and +1.5, marked shaded area)*

[Figure]

*Figure S 4 (Continue). Modeled bias in average precipitation in each dataset*

[Figure]

*Figure S 4 (Continue). Modeled bias in average precipitation in each dataset*

[Figure]

*Figure S 4 (Continue). Modeled bias in average precipitation in each dataset*

[Figure]

Figure S 5. Modeled mean trends marked the periods experiencing significant changes (upper), the first derivatives of modeled mean (lower).

[Figure]

*Figure S 5 (Continue). Modeled mean trends marked the periods experiencing significant changes (upper), the first derivatives of modeled mean (lower).*

---

## Author Response (AR1)

**Reviewer #1:**

**General comments**

**This manuscript presents a description of the model, analysis, and research gap. The writing and derived conclusion are clear. Some assumptions need further analysis to be expressed correctly. The graphs and tables provide valuable information for the analysis but deeper interpretations are needed. The dataset provides multiple ways to formulate the research approach that can be further explored.**

We appreciate the thoughtful reviews from the three reviewers and have made every effort to address each of their comments. Our responses are shown in bold below and correspond with a manuscript with edits shown. All line numbers refer to the document with tracked changes.

**Major comments**

**Line 146. The window of a 3-month moving average is not small? considering tree ring only provide growing/not growing period?**

> We focused on measuring seasonality in drought severity and its changes. The 3-month average precipitation (SPI-3) is suitable for capturing seasonal characteristics as previous studies have already demonstrated (Patel et al., 2007). Additionally, it aligned with the primary tree growth season in the underlying NASPA tree-ring reconstruction, which covers the 3-month May-June-July data.

**Line 147. The sixth percentile is a bit vague, even though it is a fitted two-parameter Gamma distribution. What is the difference with a 5%-10% threshold for the results?**

> We defined the $6^{th}$ percentile threshold based on the drought severity at SPI = -1.5, which is used in McKee et al., 1993 to define the threshold between severe and moderate drought (Table 1). As defined by the normal distribution used in the SPI calculation, the probability of SPI = -1.5 is 0.067.

> This threshold can be changed based on the severity of drought we are interested in. Similar results are expected if the threshold is modified to the $5^{th}$ to $10^{th}$ percentile range, representing SPI values between -1.64 and -1.28, which are near the boundaries of severe – moderate drought according to McKee's original categorization.

*Table 1. Meteorological Drought Categories presented. (McKee et al., 1993)*

| SPI Values | Drought Category |
|---|---|
| 0 to -0.99 | Mild drought |
| -1.00 to -1.49 | Moderate drought |
| -1.50 to -1.99 | Severe drought |
| ≤ -2.00 | Extreme drought |

**Line 187. Does the assumption of a better non-linear fitness than linearity come from drought trends' particular behavior? The obtention of the smoothed long-term trend is the reason? (Line 195).**

The drought trends have shown complex temporal changes in the real world (Ge et al., 2016). Observations suggest that climate systems including drought and precipitation have not been monotonic, but non-linear under a non-stationary climate, undergo a gradual change in variance/extremes, or both (Pryor et al., 2009; Stagge et al., 2017; Wu et al., 2017). As such, simple linear regression methods cannot capture non-linear patterns, such as a "pause", acceleration in the rate of change, or abrupt changes. For example, a non-linear trend is necessary to capture a relatively stable climate during pre-Industrial conditions, followed by more rapid climate change during the Industrial period. Assuming a linear trend over the last millennium would flatten this behavior. Additionally, the model structure, separate from non-linearity, allows for independent trends in the variance and central tendency, permitting negligible changes in the mean, but significant changes in the extremes. For these reasons, we analyzed non-linear trends within a GAM structure.

**Line 232. Which is the adequate NASPA reconstruction skill threshold and why? Table 1, shows a lower score of 0.220.**

We have added text to clarify this (Line 261). We recommend avoiding a fixed threshold for reconstruction $R^2$ between acceptable and not. Instead, we include the information in Table 1 to clarify which seasons are best for various regions.

Line 294:

NASPA reconstruction skills are investigated via calibration and validation statistics by data creators (Stahle et al., 2020). One of the calibration statistics, the coefficient of multiple determination (R2), is presented in Table 1. We avoid determining whether the datasets are acceptable or not through these statistics, rather include to clarify which seasons or regions have better skills.

**Line 369. How much can be considered natural variability?**

Separating the impact of natural variability from anthropogenic activity requires a further extensive attribution study, and is outside of our objectives in this first study demonstrating a new quantitative methodology (Deser et al., 2012; Swanson et al., 2009). Although we do not explicitly quantify or attribute the underlying mechanism of anthropogenic impact, it is reasonable to presume that anthropogenic impacts exist if monotonic onset during the 20[th] century consistently dominate trends, especially if these trends do not appear during the pre-Industrial period (Swanson et al., 2009). In that sense, our demonstration in **Line 462** "other seasons have

experienced what is **presumed** to be the natural variability **with no abrupt 20th century changes"** is a reasonable approach to interpret our results.

**Line 393. Natural variability should be addressed. Also, regarding altered seasonality alteration, is it in the natural variability to state that is drier or wet?**

The altered seasonality in the Wax, TX is caused by the combination of wetting MJJ and no dramatic change in other seasons. As such, this region had a relatively wet cool season (NDJ, FMA) and dry warm season (MJJ, ASO) until the 1900s, but is shifting to a longer and more consistent wet season (NDJ, FMA, MJJ) with a dry ASO season. Authors would like to highlight that we cannot statistically attribute whether this seasonal shift is part of natural variability or anthropogenic climate change. However, what we would like to demonstrate here is that 1) the average trend in MJJ has been gradually wetting since 1000 CE, while 2) the trends in the other seasons have been consistent for the last 2000 years. As mentioned previously, consistent climate over 2000 years might be the part of natural variability as no abrupt changes have been detected in recent years.

We have added phrase for more clarification in Line 463:

Seasonal shifts also appeared in Wax, TX, stemming from a constant wetting trend of MJJ season since 1000 CE while other seasons have experienced what is presumed to be natural variability with no abrupt 20th century changes, although further analysis is required to quantify the impact of natural variability.

**Minor comments**

**Line 103. Does the application of KNN for downscaling make it a novelty itself, just because it was not performed in tree-ring data?**

Thank you for bringing up this concern. The authors highlighted the novelty of a multi-step process that encompasses temporal downscaling, bias correction, and non-linear trend modeling. Regarding the sentence you mentioned, we agree that it requires further clarification. We have revised the sentence to label the method as the "K-nearest neighbor method" instead of referring it as "novel method" in Line 119:

Original: Section 2.3 introduces *the novel approach* for temporal downscaling of the reconstructed precipitation.

Revised: Section 2.3 introduces ***the K-nearest neighbor (KNN)*** *approach* for temporal downscaling of the reconstructed precipitation.

**Line 138. Is it correct to use the n-days distribution to convert accumulated precipitation into the standard normal distribution?**

This is the generally accepted approach used by most drought monitoring organizations for normalizing the SPI. While one could disagree with the use of the SPI, it has become ubiquitous, and an argument against its use is outside the scope of this study.

**Line 203. Why the parameter's shape can change slowly on a multi-decadal scale?**

The shape parameter of the Gamma distribution governs the shape of the probability affecting all moments, including the Mean (shape × scale), Variance (shape × scale$^2$), Skew, and Kurtosis. The skewness and kurtosis of the Gamma distribution only depend on the shape parameters, determining the behavior near zero and in the positive tail.

In this regard, a slowly changing shape parameter in our model is necessary to capture the time-varying increases in variance and tail behavior (extremes) suggested by prior climate change studies. For example, while the mean could remain consistent across centuries, a changing shape parameter could produce a shift from lower inter-annual variance to a distribution with a heavier right tail associated with more extreme precipitation events and larger variance, (Shiau, 2020; Stagge and Sung, 2022).

**Figure 5. The SPI +1.5 to -1.5 is only different for the ASO. Is it better to provide another visualization?**

The precipitation ranges between SPI = +1.5 and -1.5 are represented as grey shaded area, and the figure distinguishes this area for each dataset. For more information, we have included additional figures in Supplement Fig S4, demonstrating the precipitation distributions for all locations with distinctive color indicators for each dataset. We do believe these additional figures will help to clarify the results and provide more context, though their inclusion in the main body of the article would be excessive. We include a reference to this supplement (Line 300) together with Fig 5.

**Line 327. Which is the order of magnitude improvements of the dsNASPA (of negative bias) over drought severity?**

We did not quantify the exact magnitude of improvement, however, our modeling approach identified biases in both dsNASPA and CRU datasets, and calibrates its modeled mean based on GridMET while preserving its common gradual trend.

To elaborate, our findings presented in Figs. 5 and 6 indicate that the modeled mean of dsNASPA is adjusted upwards from its original value. This adjustment is the result of bias correction, which was employed to align the dsNASPA climatology (was slightly drier) with the wetter GridMET climatology for this season and location, Monte Vista, CO (Fig. 5).

**Line 344. Why is the drier trend especially severe? Is it the only one with this trend on the west coast? usual is the current California drought in the context of the last millennium? While drought conditions are expected to be exacerbated as a consequence of anthropogenic warming and increased evaporative demand**

We initially intended "severe" to refer to the severity of the modern (20$^{th}$ century) drying trend, rather than a comparison with the drying trend from other gauges in the region. We have clarified this sentence to make this clearer. While this appears to be at least partially an anthropogenic effect (intensifying in the Industrial period), more work is needed to quantify these drivers, e.g. (Griffin and Anchukaitis, 2014). Even more research using gridded data would be required to fully define the spatial extent of this trend, which this group is already developing.

Revised text (Lines 437 – 439):

The drier trend in Los, CA during FMA precipitation has declined since 1500 CE but this trend was exacerbated and became more severe during the 20th century, effectively shortening the winter wet period prior to the region's dry summer.

**Line 378. Is wider uncertainty the correct way to refer to higher uncertainty?**

The authors agree with your comments, we fixed it to *higher* uncertainty.

**Line 425. Improve writing.**

The sentence is revised as follows in Line 545:

Original: We make our data publicly available to allow researchers to access for develop drought trend studies.

Revised: In accordance with FAIR data principles, we make our data publicly available to allow researchers to access *it and develop future drought trend studies* (Wilkinson et al., 2016)*.*

**References**

Deser, C., Knutti, R., Solomon, S., and Phillips, A. S.: Communication of the role of natural variability in future North American climate, Nat. Clim. Change, 2, 775–779, https://doi.org/10.1038/nclimate1562, 2012.

Ge, Y., Apurv, T., and Cai, X.: Spatial and temporal patterns of drought in the Continental U.S. during the past century, Geophys. Res. Lett., 43, 6294–6303, https://doi.org/10.1002/2016GL069660, 2016.

Griffin, D. and Anchukaitis, K. J.: How unusual is the 2012–2014 California drought?, Geophys. Res. Lett., 41, 9017–9023, https://doi.org/10.1002/2014GL062433, 2014.

McKee, T. B., Doesken, N. J., and Kleist, J.: THE RELATIONSHIP OF DROUGHT FREQUENCY AND DURATION TO TIME SCALES, Am. Meteorol. Soc., 179–184, 1993.

Patel, N. R., Chopra, P., and Dadhwal, V. K.: Analyzing spatial patterns of meteorological drought using standardized precipitation index, Meteorol. Appl., 14, 329–336, https://doi.org/10.1002/met.33, 2007.

Pryor, S. C., Howe, J. A., and Kunkel, K. E.: How spatially coherent and statistically robust are temporal changes in extreme precipitation in the contiguous USA?, Int. J. Climatol., 29, 31–45, https://doi.org/10.1002/joc.1696, 2009.

Shiau, J.-T.: Effects of Gamma-Distribution Variations on SPI-Based Stationary and Nonstationary Drought Analyses, Water Resour. Manag., 34, 2081–2095, https://doi.org/10.1007/s11269-020-02548-x, 2020.

Stagge, J. H. and Sung, K.: A Non-stationary Standardized Precipitation Index (NSPI) using Bayesian Splines, J. Appl. Meteorol. Climatol., 1, https://doi.org/10.1175/JAMC-D-21-0244.1, 2022.

Stagge, J. H., Kingston, D. G., Tallaksen, L. M., and Hannah, D. M.: Observed drought indices show increasing divergence across Europe, Sci. Rep., 7, 1–10, https://doi.org/10.1038/s41598-017-14283-2, 2017.

Stahle, D. W., Cook, E. R., Burnette, D. J., Torbenson, M. C. A., Howard, I. M., Griffin, D., Diaz, J. V., Cook, B. I., Williams, A. P., Watson, E., Sauchyn, D. J., Pederson, N., Woodhouse, C. A., Pederson, G. T., Meko, D., Coulthard, B., and Crawford, C. J.: Dynamics, Variability, and Change in Seasonal Precipitation Reconstructions for North America, J. Clim., 33, 3173–3195, https://doi.org/10.1175/JCLI-D-19-0270.1, 2020.

Swanson, K. L., Sugihara, G., and Tsonis, A. A.: Long-term natural variability and 20th century climate change, Proc. Natl. Acad. Sci., 106, 16120–16123, https://doi.org/10.1073/pnas.0908699106, 2009.

Wilkinson, M. D., Dumontier, M., Aalbersberg, Ij. J., Appleton, G., Axton, M., Baak, A., Blomberg, N., Boiten, J.-W., da Silva Santos, L. B., Bourne, P. E., Bouwman, J., Brookes, A. J., Clark, T., Crosas, M., Dillo, I., Dumon, O., Edmunds, S., Evelo, C. T., Finkers, R., Gonzalez-Beltran, A., Gray, A. J. G., Groth, P., Goble, C., Grethe, J. S., Heringa, J., 't Hoen, P. A. C., Hooft, R., Kuhn, T., Kok, R., Kok, J., Lusher, S. J., Martone, M. E., Mons, A., Packer, A. L., Persson, B., Rocca-Serra, P., Roos, M., van Schaik, R., Sansone, S.-A., Schultes, E., Sengstag, T., Slater, T., Strawn, G., Swertz, M. A., Thompson, M., van der Lei, J., van Mulligen, E., Velterop, J., Waagmeester, A., Wittenburg, P., Wolstencroft, K., Zhao, J., and Mons, B.: The FAIR Guiding Principles

for scientific data management and stewardship, Sci. Data, 3, 160018, https://doi.org/10.1038/sdata.2016.18, 2016.

Wu, J., Chen, X., Yao, H., Gao, L., Chen, Y., and Liu, M.: Non-linear relationship of hydrological drought responding to meteorological drought and impact of a large reservoir, J. Hydrol., 551, 495–507, https://doi.org/10.1016/j.jhydrol.2017.06.029, 2017.

**Reviewer #2**

**Submitted to HESS**

**The paper aims to assess recent trends in seasonal precipitation in the USA in a long-term climatological context. For this purpose, a novel temporal reconstruction or downscaling approach is presented, to derive precipitation (3-mon aggregate) at a monthly time-step from tree-ring based reconstructions at a seasonal time step. Such downscaling is of general interest for climate change assessment in hydrology and meteorology and fits well within the scope of HESS.**

**The paper is generally well written and organised. However, I have some comments that include methodological concerns that should be considered before publication.**

Thank you for your thoughtful review and comments. We have sought to address your comments and suggestions below. The marked line numbers follow the article with tracked changes.

**Methodological concerns:**

**A) Temporal downscaling using K nearest neighbour resampling**

**1) The methods take into account matching 13-month time segments of the past based on three "SPI matching points", regardless of the season. However, precipitation has a typical seasonal regime, and I would strongly assume that a seasonal-correct matching would be essential to extract realistic monthly SPI (or precipitation) scenarios from the past.**

We appreciate the reviewer's concern about seasonal-correct matching potentially better capturing some characteristics of the seasonal regime beyond the probability distribution, e.g. temporal autocorrelation or sudden onset changes aligned with seasons. We considered this issue when originally designing our approach, but ultimately were forced to balance the benefits of a larger library of feasible SPI traces against the risk of missing some seasonally-specific time series behavior. We chose the former, with the additional assumption that anchoring the time series behavior at three seasonal points within the year would likely oversample sequences with similar seasonal behaviors. We have added text to the discussion in the manuscript to explain this design decision and acknowledge that more work could be done in future research to explore the magnitude of seasonality effects on SPI sequences.

Lines 475 - 482:

When developing the KNN approach, we chose to consider all 13-month time segments, regardless of seasonality, which may not capture some higher order characteristics like seasonal correlation. This design decision was a trade-off between the benefits of a larger sampling library of feasible SPI traces and the risk of overlooking some seasonally-specific time series behavior. We chose the former, with an additional assumption that anchoring the time series behavior at

three seasonal points would likely oversample segments with similar seasonal behaviors. Also, our process of selecting SPI sequences and converting back to precipitation based on the seasonal probability distribution reflects the region's seasonal characteristics. This is demonstrated in Fig. S1 showing that our dsNASPA captures the general seasonality well. Still, future research might explore the magnitude of seasonality effects and persistence on SPI sequences in the downscaling process.

**2) Calculating a mean SPI pattern as a mean of SPI scenarios is not fully correct as SPI values are highly nonlinear transformations of monthly precipitation and therefore not additive. I suggest either calculating average monthly precipitation directly from the precipitation series, or to analyse the sensitivity of your approach with regard to the correct calculation.**

Our method aligns with your suggestion: directly calculate the precipitation of each of the 10 neighbors based on the gamma distribution and then average these 10 precipitation values before transforming into an SPI value. We fully agree that the SPI values are non-linear and non-additive. As we acknowledge that the description in the downscaling process may have lacked clarity, we have revised to improve explanation as follow in lines 197-199:

Revised: Then, the ten monthly resampled SPI-3 time series were converted back to the 3-months precipitation using 2-parameter Gamma distributions derived from the GPCC dataset. Lastly, the 10 sets of precipitation time series were averaged and inserted into the targeted year of the NASPA.

**3) Using the average temporal signal will filter out climate (i.e. between-year, or event-specific) variability of events, which restricts the possible user value of the reconstructions data set. I would appreciate to clarify this in the aims and add to the discussion. From my point of view, the strength of the ensemble of history-based scenarios over some "average" scenario would be that the ensemble represents the diversity of realistic realizations conditional to the given information (three matching points), while the mean is not thus realistic (as it never happened) and unrealistically smooth, too.**

We have added the following sentences in discussion in lines 492-499.

Our primary objective is a realization of the best possible estimate of the changes in precipitation distributions (climatology) of the past, rather than to replicate specific events in the time series. If one was interested in predicting precipitation in a given year, we recommend using the original NASPA dataset, rather than our dsNASPA interpolation. Overall, our derived Gamma distribution can be used to understand the most likely climatology of the past, and potentially, the future based on available data.

In this context, our KNN approach creates a plausible estimate for periods lacking NASPA estimates (e.g., ASO) and is apt for estimating smooth changes in the distribution of historical

climatology. As further support, we found that taking the mean of nearest neighbor ensembles did not artificially and dramatically decrease the variance (Fig. S1).

**B) Bias correction**

**When the ultimate aim of this paper is to analyse nonlinear long-term trend behaviour, I wonder how far bias-correction will have any effect on the results. Did you correct only biases in the mean or also in the shape parameter? Section 3.2 suggests that both are corrected, but e.g. Fig. 6b ASO (cp. to Fig. 5) suggests that this is not the case. There is also a strong bias in the long-term average visible in Fig. 6b NDJ.**

Thanks for your comments. As mentioned in the text (Lines 320), both shape and mean parameters are corrected. This is illustrated as an adjustment in precipitation values at mean (SPI= 0), and SPI = +1.5 and -1.5 (0.068 and 0.933 percentile) in our results. For example, precipitation at SPI= +1.5 of NASPA (Fig. 5. ASO) is computed around 40 mm/month but is corrected as 60mm/month when it is adjusted using Gridmet (Fig. 6b, ASO).

As you correctly note, it is possible to see a correction for both the mean and shape in Fig 5, particularly for where the NASPA overly compressed the variance (Lines 330-332). Fig. 5 was intended to demonstrate the bias correction process, whereas Fig. 6 was intended to emphasize the trend in the SPI=0, -1.5, and +1.5 over the last 600 years, after bias correcting using Gridmet as a baseline. Fig 6 only displays the post-bias correction common trend for clarity. We have added the following text to the caption of Fig 6:

Trends in the mean and SPI = -1.5 to +1.5 range are shown using Gridmet as baseline to illustrate bias correction, while the raw data is shown without bias correction for context.

**Further, I would assume that bias correction (in any of the parameters) will only change the results (incl. quantile values) if it is performed over the whole series in a non-constant way, and I wonder how this can be performed for the longer past before the CRU and GridMET observation periods.**

We assume bias in the mean and shape parameters, determined during the overlapping data period, remains constant in the long-term (See Lines 263). This assumption is necessary for any bias correction to function outside the common observed period and is the same principle that underlies delta index or quantile matching bias correction for climate model projections. Based on NASPA validation, we feel reasonably confident that bias remains consistent during the period with consistent tree-ring coverage (1400-present, shaded white in Fig. 7), but may begin to change as chronology coverage and reconstruction skill decreases (beyond 1400, shaded grey in Fig. 7). We therefore chose to show the full period reported by the NASPA but recommend caution for the most distant periods (Lines 428 - 429).

**Specific comments:**

**The aims of the study and methodology need to be sharpened in the text.**

**"This study is designed to address the challenge of merging precipitation datasets with varied biases and temporal scales to calculate a common Standardized Precipitation Index (SPI; Guttman, 1999) meteorological drought series that incorporates non-stationarity" (L. 84) seems not to fit to the actual study where the target is a downsclaed monthly precipitation series (dsNASPA, Fig. 1) and finally a 3-month precipitation average (L. 170).**

We revised lines 84-85 as follow:

This study is designed to address the challenge of constructing 2000 years of precipitation climatology by merging multiple datasets with varied biases and temporal scales.

**Similarly, with BIAS correction: I had a hard time to understand whether the study really aims at bias-corrected dsNASPA (as clearly stated in Section 2.4), or just investigating model bias (Sect. 3.2) and later constructing long-term trends (Sect. 3.3). (But I think this can easily be clarified.)**

We have revised lines 86-92 as follows:

The objectives of this study are therefore to (1) construct downscaled NASPA precipitation time series from bi-annual into monthly scale with 3 months average resolution, (2) identify unique biases inherent in different precipitation data and remove those biases, and ultimately (3) construct a 2000 year continuous climatology model that can capture century-scale shifts in the 3-month precipitation. This approach mimics the underlying distribution methodology of the Standard Precipitation Index. The continuous climatology derived from proxy reconstructions and modern observations is the true goal, with the first two objectives functioning as necessary intermediate steps towards this ultimate goal.

**2.3: KNN approach**

**L. 157: I think the aim here is not to simulate uncertainty, but to cover the range of possible monthly precipitation patterns behind the 3 matching points. As you have absolutely no information what was in between the 3 matching points, all "well matching" patterns are equally likely (same for L. 179).**

We agree with your comment and have revised the text. Also, we have removed mentioning the resampling approach as per your comments in PDF file in this section (the comment was: *Would not call this a resampling approach, it is rather selecting history-based scenarios*). Please refer below:

Line 180:

Original: Because this is a resampling approach, multiple (K = 10) annual historical sequences are inserted for each year of the reconstruction to simulate uncertainty.

Revised: To do this, multiple (K = 10) annual historical sequences are inserted for each year of the reconstruction to approximate plausible monthly precipitation patterns that most closely match the three NASPA reconstructed periods.

Line 204 - 206:

Original: Third, the K neighbors create an ensemble of time series, incorporating uncertainty.

Revised: Third, the K neighbors create an ensemble of equally likely time series, identifying an envelope of feasible time series when there is no information between the 3 points from the NASPA reconstruction.

**The KNN could be directly based on monthly precipitation rather than on SPI (see general comments).**

Please refer to our response above.

**L. 178: "Second, direct resampling based on similarity from the GPCC sample field incorporates realistic seasonal progression and persistence of the SPI" – Only if just the seasonally aligned patterns are used (see my previous comment).**

As mentioned above, our SPI nearest neighbors provide plausible patterns of "anomalies". Converting the "anomaly" back to precipitation based on the seasonal probability distribution provides realistic seasonality. We agree that including "persistence" in this statement without clarification may have been too broad. We originally meant that resampling from historic GPCC SPI-3 sequences would incorporate the structural persistence from the 3-month moving average process, which contributes the majority of persistence (temporal autocorrelation). But, we acknowledge that ignoring seasonality in resampling was a design decision that may miss some seasonal-specific persistence due to atmospheric circulation (see comment for Methodological Concern A). We have therefore modified this sentence to read "… realistic seasonal progression and the 3-month structural persistence of the SPI-3" (Line 204). We have also added discussion about the benefits of using seasonally aligned samples and suggested future study in Line 475-482.

**2.4: Section is hard to follow, suggest to improve clarity.**

We have revised this section to improve clarity and addressed all the detailed comments below and in the attached file.

**Please state at the beginning the aim of this step (e.g. Is it adjusting your NASPA reconstruction to long-term fluctuations and average seasonal cycle of one/two different precipitation data sets? Or is it just estimating the temporally varying parameters of the Gamma distribution to be used for SPI calculation – so is it really bias-correction then?)**

Thanks for the suggestion. The aim of the GAM modeling has been added in lines 216 - 221:

This method was applied to create a single, common estimate of the temporally varying Gamma distribution parameters representing precipitation climatology by incorporating information from multiple biased data products. We refer to the process of accounting for seasonal bias in the mean and shape parameters from different data sets as "bias correction" for the remainder of this paper because it mirrors the process of bias correction by moment matching. However, unlike a separate bias correction step, this is performed within the GAM model, permitting confidence intervals around each of the bias correction terms.

**The presentation of the model (Eq. 1 and 2, text around) is confusing. All parameters need to be explained in the text, consistent use of beta or B (L.206), redundant use of beta for model parameters of both Gamma parameters (Eq. 1 and 2), and finally also for the scale of the Gamma distribution mu/alpha (L. 200). (See further sticky-notes in the annotated pdf).**

Thanks so much for your valuable comments. The equations were revised with distinctive notation for each model parameter and removed scale parameter notation. The model term in equation is also revised to display names of all datasets as per your comments.

$$P_{3\,month,m,y} = gamma(\mu, \alpha) \quad \begin{pmatrix} m: month\ of\ the\ year, \\ y: year \end{pmatrix}$$

$$\mu = \beta_{0\mu} \begin{pmatrix} CRU \\ NASPA \\ Gridmet \end{pmatrix} + \beta_{1\mu} f_{s\_\mu} \left( X_{month},\ by = \begin{pmatrix} CRU \\ NASPA \\ Gridmet \end{pmatrix} \right) + \beta_{2\mu} f_{te\_\mu}(X_{year}, X_{month}) \quad (1)$$

$$\frac{1}{\log(\alpha)} = \beta_{0\alpha} \begin{pmatrix} CRU \\ NASPA \\ Gridmet \end{pmatrix} + \beta_{1\alpha}\ f_{s\_\alpha} \left( X_{month}, by = \begin{pmatrix} CRU \\ NASPA \\ Gridmet \end{pmatrix} \right) + \beta_{2\alpha}\ f_{te\_\alpha}(X_{year}, X_{month}) \quad (2)$$

where $P_{3\,month,m.y}$ represents the 3-month moving average precipitation at year $y$ and month $m$. The precipitation is fitted in Gamma probability distribution, where have $\mu$ (mean) and $\alpha$ (shape parameter). The $\beta$s are different parameters of each spline function, $f_s$ and $f_{te}$, which denote cyclic and tensor spline, respectively. The underlying principle of the model is that there exists a single best estimate of the precipitation distribution at any given time, described by the mean and shape parameters of the gamma distribution that changes seasonally $\beta_1 f_s(X_{month,}by = dataset)$ and can also change slowly on a multi-decadal scale, $f_{te}(X_{year}, X_{month})$ with constant y intercept, $\beta_0(dataset)$.

**Can you be more specific about the GAM model components? For the long-term component, what is the nature of the spline, what is the time-window of smoothing, how has it been chosen/optimised? And: how does it behave in the observation period of two, one and finally no instrumental records, and what is the nature of bias-correction in the three periods, esp. in the pre-instrumental period?**

We applied the tensor spline as a long-term component, enabling us to model interactions of multiple variables in different time units. The smoothed is constrained every 70 years to mimic changes in multi-decadal scale with preventing overfitting.

This is added in the text as below in main text lines 251 - 257:

The single common tensor product spline smoother ($f_{te}(X_{year}, X_{month})$) is shared across all datasets to model the interaction of long-term trends ($X_{year}$) relative to season ($X_{month}$) using smoothly changing parameters for the two dimensions (year and month). A tensor product spline is an anisotropic multi-dimensional smoother, meaning it can model the interaction of variables with different units and can assign different degrees of smoothing for each direction, as is necessary for dimensions of month and year. Estimating $\beta_{2\mu}$ and $\beta_{2\alpha}$ in terms of year and month allows for non-linear annual trends for each month while constraining these trends to be smooth through time. We constrain the smoother with control points (knots) every 70 years for mean and shape parameters to approximate climate variability on decadal scales while preventing excessive sensitivity/volatility.

**Further, L. 195 "a smooth long-term trend that is common to all data sets (CRU, GridMET, and NASPA) – reads like a contradiction to L. 223: "Following the first assumption, … we adjust CRU and NASPA parameters to match the GridMET dataset"**

We have changed the first sentence as follows based on another comment for clarification (Line 231-235):

When applied, this model decomposes information from all datasets (CRU, GridMET, and NASPA) into a smoothed long-term trend that is common to all datasets and an additional annual seasonality smoother that varies slightly by dataset to account for bias relative to GridMET. In this way, there is a single common trend, with an adjustment added to shift the mean and shape parameters up or down seasonally based on the data source.

**And: What is the model vs. the data set? (If this is the same, than please use one term consistently)**

The term "model" in the main text is all revised, and the equations are revised to display names of all datasets instead of referring "model".

For instance,

Lines 261 - 263: The first two terms derive intercept and seasonality distinctive to each dataset. The first term ($\beta_0 \ (datasets)$) accounts for dataset specific intercept and the second term ($\beta_1 f_s(X_{month}, \text{by} = dataset)$) accounts for dataset specific seasonality.

**3. Results**

**3.1 & Fig.1: Are these all bias-corrected values or not? Which values need to be bias-corrected, all apart from GridMET to make them comparable to GridMET?**

Figure 1 and Section 3.1 refer only to pre-processing the NASPA data (temporal downscaling to fill gaps) prior to the GAM model that accounts for bias. Therefore, all the values are before bias correction. You are correct, those values are not related to GridMET. To make them comparable to GridMET, they are then bias corrected in the next step (Sections 2.4 and 3.2).

**L. 258: "Despite the limitation, downscaling accurately predicts the general precipitation pattern in terms of seasonal and long-term average precipitation, with nMAE values generally between 0.1 - 0.5.": Can you compare this with a naïve estimator, being the long-term monthly average precipitation (highly naïve), or better the monthly precipitation quantile values qgam (alpha), where probability alpha corresponds to the anomaly linearly interpolated between the matching points? How much is the gain in performance using your reconstruction (downscaling) approach?**

This is a reasonable request to provide context for the nMAE. We have calculated nMAE for the highly naïve estimator, ignoring NASPA reconstructions and instead using mean climatology. The figure below compares the nMAE of our dsNASPA and that of highly naïve estimator. We found dsNASPA provides more accurate signal in months with some NASPA information (December through July), as one would expect, while providing approximately similar signal in the months without NASPA information. It should be noted that the dsNASPA interpolation scheme prevents extremely high errors (nMAE > 1, off the y-axis scale) in certain regions of the semi-arid Western US where NASPA has the most skill. Fig. S3 further demonstrates the validity of our results in each season, not only with respect to mean absolute error, but with visible correlation between observations and dsNASPA. A naïve estimate would produce a single estimate (vertical line) with none of this correlation. We are confident that all three seasons except for ASO capture enough long-term characteristics to investigate smooth trends in each season. We have added two sentences explaining the performance of dsNASPA compared to this very simplified model.

Line 329-334: We compared performance of the dsNASPA with a highly naive alternative (assuming the mean GPCC climatology) and found that dsNASPA provides a clear signal in the period with NASPA information (blue shaded period in Fig 4). As expected, the dsNASPA provides less information in the interpolation period where NASPA estimates are not available. However, during these gap seasons, the dsNASPA still produces positive correlation with observations, useful for measuring climatological shifts, and greatly reduces extreme errors created by the naïve estimator in the semi-arid West.

We initially considered a linear interpolation approach, but excluded it because it provided only a single estimate for each time step, essentially assuming the same uncertainty whether we are at a time step with a NASPA estimate (e.g. MJJ) or if we are interpolating between NASPA

estimates (e.g. ASO). This means we could not produce the envelope of feasible paths between the three NASPA anchor points, with the envelope naturally getting wider as we moved further from the NASPA estimates in time and tightening as we approached each estimate. We considered this information critical and therefore discounted the linear interpolation approach.

[Figure]

Figure 1. Comparison of nMAE for each season, showing. (Left) nMAE of dsNASPA (Left and reproduced from Fig. 4) and nMAE for the naïve long-term average estimator (Right). Note, the y-axis is mirrored for the naïve estimator, which cuts off some of the worst nMAE scores (> 1 in much of the semi-arid West).

**3.2 Investigating model bias**

**REM: Here I believed to see that you are not performing bias correction, but you rather use the GAM model to decompose the 3 signals (dsNASPA, CRU, GridMET) into the long-term trend and seasonal average components (please make this clear in intro and methods section). Pls. clarify.**

We revised the introduction to highlight how to account for bias through modeling in lines 96-98:

Original: Second, we develop a model to simultaneously analyze unique biases in proxy and instrumental datasets, account for them, and then analyze a non-linear trend common across the datasets.

Revised:  Then, we develop a model to simultaneously capture non-linear trends while accounting for unique biases across proxy and instrumental datasets by decomposing information from all datasets into their shared long-term trends, seasonality, and data-specific bias.

Also, our method section (lines 229 - 235) demonstrates how our GAM model works in analyzing data-unique seasonality and shared long-term trend. We added more explanation based on your suggestion:

Original: Here, we expand this approach, by adding a hierarchical grouping variable to simultaneously model common seasonal-specific long-term trends across datasets, while

incorporating variability at the group level following the approach of Pederson et al. (2019). When applied to our model, this involves a smoothed long-term trend that is common to all datasets (CRU, GridMET, and NASPA) and differences for the seasonal terms that repeat annually at the group-level (climate dataset) (Pedersen et al., 2019). The modeled data-unique seasonality can capture seasonal bias in each dataset, and it was further adjusted to construct the long-term trend time series.

Revised: Here, we expand this approach, by adding a hierarchical grouping variable to simultaneously model common seasonal-specific long-term trends across datasets, while incorporating variability at the group level following the approach of Pederson et al. (2019). When applied, this model decomposes information from all datasets (CRU, GridMET, and NASPA) into a smoothed long-term trend that is common to all datasets and also an annual seasonality that varies slightly by dataset to account for bias relative to GridMET. In this way, there is a single common trend, with an adjustment added to shift the mean and shape parameters up or down seasonally based on the data source.

**This section is well written (pls. use consistent names of the three monthly products)**

We refer to the name as 3-month precipitation.

**The bias in the long-term trend component in the particular station shows quite significant underestimation. Can you generalize this finding to all 14 stations?**

Thanks for the suggestion, we have added additional figures in the supplement (Fig. S4) to show bias in four equally spaced seasons at all study sites. We believe that these new figures will improve understanding of the bias inherent in three datasets in our study sites and provide complete transparency for any reader who is interested.

Line 370-376:

We present the results for all other regions in Fig. S4. The results indicate that the shape biases are largely dependent on the season, whereas mean biases are more dependent on the gauge. Notably, the ASO season shows large biases in the shape parameter. This is primarily because the dsNASPA in this season can't represent occasional extreme precipitation values, inducing an underestimation of its variance. In contrast, the MJJ season shows considerably less bias since the dsNASPA has developed from complete precipitation information in NASPA. A few exceptions exist in Mtv, CO and Grd, MT, where have large biases in the mean parameters across all seasons, possibly due to topographic effects between the gauge locations in these mountainous regions.

**3.3 Constructing long-term trends**

**REM: Here I understood that in a second step you now indeed correct for all biases assessed in the step before.**

**"By accounting for the model-induced bias described in Section 3.2 and adjusting all datasets to match GridMET, we were able to generate a 2000-year model of non-stationary precipitation trends for each gauge."**

**Given the nonlinear nature of long-term trend models, how much credence do we have that biases found in the short instrumental periods (~40 and ~120 years), by a model forced to parallel log-term trend components, can be generalized to the past 2 millennia? (This should be discussed, e.g. also in the light of the finding of Duethmann & Blöschl (2020) about instrumental biases in precipitation including station density).**

> Thank you for raising this concern. We agree that assuming stationarity in bias is a caveat of this method. However, it is a necessity that underlies nearly all proxy reconstructions. We have added text to highlight this caveat and to reiterate our much stronger confidence in the stationarity of bias back to 1400 (illustrated in Fig 7) and discussed in our response to Methodological Concern B.
>
> Line 487 - 490:
>
> Stationarity in bias is an assumption of this method, however, it is a necessity that underlies most proxy reconstructions. Based on prior NASPA validation, we are most confident that bias remains consistent during the period with consistent tree-ring coverage (1400-present, shaded white in Fig. 7), but may begin to change as chronology coverage and reconstruction skill decreases (beyond 1400, shaded grey in Fig. 7).

**Looking at the plots, it seems that only biases in the mean long-term trend and not in the shape/spread parameter are corrected? Please clarify.**

> The bias correction in shape parameter can be observed by the intervals between precipitation values at SPI = +1.5 and -1.5. For example, in Figure 5 ASO season, the dsNASPA shows a relatively small interval between SPI = + 1.5 and -1.5 compared to Gridmet dataset. Our model adjusts the shape parameter of dsNASPA based on Gridmet, resulting in larger interval as shown in Figure 6b, ASO. The result indicates that the modeled wet anomaly (at SPI = +1.5) is far outside of the dsNASPA variability as the y-intercept and seasonally varied bias in shape parameters are corrected and have continuous smoothed ranges in line with Gridmet.

**Further, please clarify if all the statements about wet and dry periods, etc. in this section are derived from the dsNASPA, or from the other series.**

Our statements are based on the common long-term signal from all data. You are correct that beyond 1900, the other data is not available and so the model collapses to be informed by dNASPA. Other estimates of pre-Industrial climate could be incorporated into this framework in the future.

**L. 304: Hard to see the described wetting/drying trends in the plot. Drying trends only in the wet anomalies but not at drought conditions – can you interpret this? Please make clear in the wording that you are reporting about wet/dry anomalies, rather than absolutely wet/dry conditions.**

We corrected the sentence as below (Line 387-389):

Original: Figure 6a shows the warm season (MJJ) has undergone a long-lasting wetting trend from the 1800s to 1900, followed by an abrupt drying trend during the 20th century. Those recent rapid drying trends are manifested in both the mean (SPI = 0) and wet condition (SPI = 1.5).

Revision: Figure 6a shows the warm season (MJJ) has undergone a long-lasting wetting trend from the 1800s to 1900, followed by an abrupt drying trend during the 20th century in both the mean (SPI = 0) and wet extreme case (SPI = 1.5).

**Fig 6a: The plot shows quite some annual differences, and biases in the spread between dsNASPA and the observation records.**

We agree. In particular, the bias in variance between observations and NASPA in FMA is an example of why bias correction term was necessary in order to assimilate the proxy reconstructions into a framework with observations. This is a particularly extreme case presented here for demonstration. You can see the remainder of similar figures in Supplement S4.

**L. 308: "The current NDJ mean is the wettest condition of the last 1000 years" – What is "current", and based on which series?**

We fixed the sentence for clarification:

The current NDJ mean is the wettest condition of the last 1000 years.

The modeled mean of NDJ in current years (2000-2020 CE) is the wettest condition of the last 1000 years (Figure 6a).

**Fig 6: red line – why is MJJ 2020 value a "modern benchmark"? (and why only for MJJ a benchmark is shown?) Further, what is the meaning of intense yellow colour at the end of FMA and ASO in panel 6a?**

Thanks for raising this issue. Our intention was to emphasize the period with notable changes, however, in response to your concern, we have removed the highlighted area and red line in MJJ 2020 to avoid confusion.

**L. 318: significant drying trend – how do you define significance?**

Thanks for your feedback. We revised the "significant drying trend" to "strong drying trend".

**L. 320: are app references in the paragraph to Fig. 6b (instead of Fig. 5b)?**

Corrected

**L. 327: Ref to Fig. 5?**

Corrected

**L. 340: "Figure 7 implies a general spatial pattern of recent drying in the western US, and wetting in the 340 eastern US." I would agree with the wetting trends in the eastern US, but do not see a general pattern of drying trends in the western US.**

We have revised as follow (Line 431 - 432):

Original: Figure 7 implies a general spatial pattern of recent drying in the western US, and wetting in the eastern US.

Revised: Figure 7 shows that the 14 demonstration  sites generally follow a spatial climate pattern found in previous studies: with Industrial-era drying trends in the southwestern US, and wetting trends in eastern US (Lehner et al., 2018; Prein et al., 2016; Ellis and Marston, 2020).

**Fig. 7: This figure gives the synthesis of the paper, by presenting seasonal trend components across the 14 stations across the US. I wonder for the methods to derive the evidence here, how far is bias-correction necessary to get evidence about wetting/drying trends as given in the text?**

Our hierarchical modeling structure adjusts the mean and shape parameters up or down using single common value and seasonally varied value, all based on GridMET dataset. Thus, our method provides reliable results in comparing long-term trends across different seasons and understanding the seasonal shifts in these trends. Also, our method makes the modeled trends in the periods with multiple available datasets are especially reliable since the shared trends are analyzed by multiple

datasets. This gives greater certainty in 20[th] century trends where we can understand modern changes.

Beyond the mean trend in Figure 7, our bias correction method effectively analyzes long-term trends in any anomalies as we adjust the shape parameters. This ensures that long-term trends at any climate anomaly during any season can be investigated, as depicted in Figure 6 and mentioned in Line 401 – 406:

The variance between SPI = 1.5 and SPI = -1.5 became much larger during the recent period, particularly for the ASO season, implying both wet and dry anomalies have become more extreme than during prior centuries. The strong drying trend in MJJ coupled with a wetting trend during the NDJ season indicates a seasonal shift of the driest season.

**Regarding the synthesis of the paper, an interesting question would be to what extent the recent trends in the three monthly precipitation series (especially stratified by season) are consistent. And given the nMAE in the FMA and MJJ season introduced by downscaling, how far the recent trends in downscaled dsNASPA and the original NASPA values coincide.**

> That is an interesting follow-up question. Our downscaling approach (essentially interpolation by resampling) and bias correction approach (essentially quantile matching) should generally maintain rank order. In other words, high anomalies in the NASPA should still be high in the dsNASPA, even with bias adjustments. So, one would expect that an increasing/decreasing trend would appear in both.

> The only case where the original NASPA and the fitted trend might disagree is if CRU and Gridmet have a completely opposite trend, in which case the model would indicate negligible trend, with a much larger uncertainty in the trend.

**Discussion:**

**Please add a discussion about the overall uncertainty of the approach. What kind of uncertainty needs to be expected given the NASPA reconstruction skill (Table 1) overlaid with downscaling uncertainty (Fig. 4)? What sources and approx. order of magnitude of uncertainty need to be expected in the longer past beyond the observation period? How shall the dsNASPA be interpreted (or: for which purposes can they be used) given the uncertainty?**

> We have added the following paragraph in line 483 - 487 in discussion: We also acknowledge the uncertainties over the whole modelling process. The uncertainties in dsNASPA stemming from the downscaling process in addition to the original reconstruction process vary by region and season. As shown in our results, the downscaling skills are much higher in the period where the original NASPA provides information (Fig. S1 and S4). Reconstruction skills in the original NASPA vary depending on the region due to the availability and sensitivity of tree-ring chronologies (Stahle et al., 2020).

Also, in Lines 491-496:

Our primary objective is a realization of the best possible estimate of the changes in precipitation distributions (climatology) of the past, rather than to replicate specific events in the time series. If one was interested in predicting precipitation in a given year, we recommend using the original NASPA dataset, rather than our dsNASPA. Overall, our derived Gamma distribution can be used to understand the most likely climatology of the past, and potentially, the future based on available data.

**Technical comments:**

**L. 241: Use term dsNASPA for downscaled (i.e. monthly) NASPA consistently throughout the text (and figures, e.g. Fig. 5).**

We fixed dsNASPA in all lines, and legends in Figure 5 and Figure 6.

**L. 148 (and throughout the text): Make clear that the SPI thresholds, values, represent wet/dry anomalies, which does not mean that the condition is absolutely dry.**

We have fixed the dry/wet condition to dry/wet anomaly or anomalies in Line 148, 343, 344, and 354

**Typos and minor comments to the wording see sticky notes in the annotated pdf of the MS (use them as appropriate).**

Thanks for your time and consideration. We have attempted to address all your comments in PDFs. Responses are attached in the sticky notes and revised text are shown in manuscript.

References:

Ellis, A. W. and Marston, M. L.: Late 1990s' cool season climate shift in eastern North America, Climatic Change, 162, 1385–1398, https://doi.org/10.1007/s10584-020-02798-z, 2020.

Lehner, F., Deser, C., Simpson, I. R., and Terray, L.: Attributing the U.S. Southwest's Recent Shift Into Drier Conditions, Geophysical Research Letters, 45, 6251–6261, https://doi.org/10.1029/2018GL078312, 2018.

Prein, A. F., Holland, G. J., Rasmussen, R. M., Clark, M. P., and Tye, M. R.: Running dry: The U.S. Southwest's drift into a drier climate state, Geophysical Research Letters, 43, 1272–1279, https://doi.org/10.1002/2015GL066727, 2016.

Stahle, D. W., Cook, E. R., Burnette, D. J., Torbenson, M. C. A., Howard, I. M., Griffin, D., Diaz, J. V., Cook, B. I., Williams, A. P., Watson, E., Sauchyn, D. J., Pederson, N., Woodhouse, C. A., Pederson, G. T., Meko, D., Coulthard, B., and Crawford, C. J.: Dynamics, Variability, and Change in Seasonal Precipitation Reconstructions for North America, Journal of Climate, 33, 3173–3195, https://doi.org/10.1175/JCLI-D-19-0270.1, 2020.

**Reviewer #3**

This manuscript introduces a numerical framework for downscaling time series of rainfall reconstructions (produced by leveraging the information contained in paleo-records). Specifically, the authors first downscale the data of the North American Seasonal Precipitation Atlas at the monthly scale, and then use the downscaled data to study long-term trends in precipitation.

I found the study to be interesting and technically sound. This said, I believe a number of aspects should be properly addressed before this work is considered for publication.

First, the quality of the presentation can be improved, especially the parts related to data and modelling approach (please refer to my detailed comments). The latter is not entirely self-contained, as it relies excessively on Stagge and Sung (2022) and Sung and Stagge (2022).

Second, the sensitivity of the results w.r.t. modelling assumptions and parameterization of the modelling framework is not explored at all. Considering the uncertainty that is associated to reconstructions (whose average skill has an R2 in the order of 0.5 or so), I would recommend the authors to identify key assumptions / parameterizations and their potential impact. Examples include the length of the moving average (3 months) or the number of neighbours in the KNN. Another key assumption that surely deserves a discussion / analysis is the one concerning the assumption that the time series of GPCC and NASPA have the same pdf.

Third, the presentation of the results could be strengthened (please refer to my detailed comments). In particular, I suggest the authors to consider the option of complementing the analysis of Section 3.3 (primarily based on the data illustrated in Figure 6 and 7) with additional quantitative analyses based on trend detection.

Thank you for sharing your comments. The authors believe that the article has been much improved by addressing your comments. We have conducted additional analysis to address your general comments as well as responding to all detailed comments. We have attached the supplement for the supplement created based on your comments.

First, Method section 2.4 was revised to better delineate the GAM modeling process. We added additional explanations and fixed the equations to avoid confusion in section 2.4.

Second, we have conducted a sensitivity analysis to assess the sensitivity of the downscaling method to the number of nearest neighbors. The performance of the KNN method was evaluated using normalized mean absolute error (nMAE) based on the GPCC time series, and the results are added in supplement figure S2. The results reveal that error (nMAE) tends to be highest for small numbers of nearest neighbors but stabilizes at 10 neighbors in the most seasons and locations, even in cases with relatively large nMAE values. It is worth noting that the dry conditions, such as JJA season in Los, CA or Phx, AZ show fairly large nMAEs due to a very small denominator in the nMAE calculation, amplifying the error values. Regardless of absolute nMAE values, 10 nearest neighbors provides stable results in predicting dsNASPA time series.

To further support the reliability of dsNASPA product, we have added additional results to demonstrate how well dsNASPA matches with GPCC observations and captures the general climate at all study locations. The results indicate that the dsNASPA captures the general seasonality and spread well for each season (Figure S1), and that the time series estimates match with GPCC observations (Figure S3, time series and scatter plots).

With regards to your concern about assuming the GPCC and NASPA sharing the same probability distribution, we would like to point out that the original process to develop the NASPA time series is based on regression using the GPCC dataset and further calibrated and validated using GPCC (Stahle et al., 2021). We are directly converting NASPA estimates back to GPCC where we have data and essentially interpolating in seasons where we have seasonal gaps. We emphasize this with our results for MJJ in Figure S3 confirming the original assumption through very well-matched time series.

Thus, we have revised the main text as below (Lines 152-154):

Original: Since the NASPA reconstructions were originally calibrated based on GPCC data, we assumed that precipitation time series of GPCC and NASPA share the same probability distribution function at the same grid cell. Thus, this study uses monthly GPCC data to best mimic the intra-annual characteristics for temporally downscaling and disaggregating NASPA estimates to monthly time series.

Revised: Since the NASPA reconstructions were originally developed at a gridded scale via regression using GPCC data, and further validated and calibrated based on GPCC data, we assumed that the GPCC and NASPA datasets share regional and temporal characteristics. Thus, this study uses monthly GPCC data to best mimic the intra-annual characteristics for temporally downscaling and disaggregating NASPA estimates to monthly time series.

Third, we have conducted a significance test for trend detection, based on the instantaneous first derivative, at all study sites (results presented in Figures S5). We took a non-linear trend analysis approach instead of simple linear trend analysis. Thus, our method overcomes the limitation of simple linear significant tests which only capture monotonic changes and are highly impacted by two separate test periods. But, in doing so, we cannot discuss a single "trend". Instead, we must discuss the trend at a given time, represented by the instantaneous first derivative. We have included a 95% confidence interval around the first derivative, to indicate periods where the trend is significantly different from zero, i.e. significantly increasing or decreasing. This method is able to preserve all non-linear and non-stationary characteristics in modeled trends and identify periods of statistically significant changes in time series.

We have added the following text in Methodology in lines 276 - 285, and results are attached in Fig. S5.

The significance of the modeled trend is tested using the instantaneous first derivative method. This method calculates the first derivative of the modeled trend using 1000 randomly drawn estimates of the modeled mean and shape parameters through time (by year). Then, we calculate the 95% confidence interval around the first derivatives to indicate periods where the trend is significantly different from zero, i.e., the trend is increasing or decreasing. The non-linear trend analysis approach overcomes the limitation of simple linear significant tests which only capture monotonic changes. In doing so, it is not possible to discuss a single "trend", but once can discuss whether the distribution mean is significantly increasing or decreasing at a given time, represented by the instantaneous first derivative. As such, this method has the benefit of preserving all non-linear and non-stationery characteristics in modeled trends, while providing estimates of significant changes. The results of this analysis are shown in Fig. S5.

**Specific comments**

**- Abstract, line 8. Perhaps too soon for a technical term like "normalized"? What does it mean here?**

We agree, it refers to drought indices which quantify drought severity using normal distribution such as SPI and SPEI. As we agree with your opinion, we removed the world "normalized".

**- Abstract, line 9. Too strong—in my opinion. How does the interpretation of drought indices produce social consequences?**

Applying different reference periods can impact on quantifying drought severity: severe drought in the current period might be a typical climate 100 years ago. We emphasized that the misunderstanding of the current situation can lead mismanagement of water resources, especially if engineered water resources systems were designed under a significantly different climatology to the one they are being applied to.

We also agreed that serious social consequences are too strong for the first sentence of the abstract. We changed this sentence in Lines 9 - 10:

Original: "impacting the interpretation of normalized drought indices and potentially having serious ecological, economic, and social consequences."

Revised: "impacting the calculation of drought indices and potentially having ecological and economic consequences"

**- Line 49-50. "from which … annual growth." Partially repeats the content of the first part of the sentence.**

We have revised the sentence as follows (lines 49-50):

For example, this study uses a reconstruction of precipitation across North America based on tree-rings, which infer the relative availability of regional precipitation or soil moisture from annual growth.

**- Line 77-78. I found this sentence difficult to follow. What does the "seasonal windows of the time series" mean? The reference to Figure 1 is not particularly useful if not combined with a decent description of the figure.**

We have revised the sentence as follows (lines 76-77):

Original: For example, the NASPA reconstructions are made up of two values per year, but the seasonal windows of the time series are of different length and do not span the entire year (Fig. 1).

Revised: For example, the NASPA reconstructions provide only two time series per year with different precipitation periods: May-July and December – April.

**- Line 85-86. "may" incorporate non-stationarity?**

We meant our calculations will incorporate non-stationarity to the typically stationary SPI.

However, we chose to remove this sentence based on another reviewer's comment.

**- Line 98. I suggest introducing acronyms (e.g., GridMET, CRU) at the right place.**

We have deleted the acronyms here to introduce in dataset section.

**- Line 101. What does "GAM" mean?**

We fixed it to show full name with acronym in Line 117: Generalized Additive Model (GAM).

**- Line 107: Why "additional"? The datasets have not been presented yet.**

Fixed. We deleted "An additional dataset," and started with "The Global Precipitation Climatology Center (GPCC)"

**- Line 116. Not clear: Where do these 16 ensembles come from? What does "regressions of differing weighs" mean? The same comment applies to Line 116-119—too many details are included without providing the right background. What is the "shared variance"? What does "DJFMA and original" mean? What does "non-processed MJJ reconstructions from the 16-ensemble" mean?**

Thank you for this comment. This is detail about the NASPA creation that is explained in far more detail in Stahle et al (2020) and is not needed here. We have simplified this section.

**- Line 126-127. I don't understand this sentence: Why do you need to "calibrate CRU and NASPA datasets"? I assume this doesn't mean that GridMET was used in the previous studies that created CTU and NSAPA, since at Line 130 you say that NASPA targeted the GPCC data.**

Thank you for bringing up this issue. We explained our methodology of using GridMET here. To clarify, we revised in Line 147-149:

Original: "The highly accurate and well-distributed precipitation data is used as the base observations to calibrate CRU and NASPA datasets."

Revised: "In this study, we assume the GridMET as a "ground truth" and use it to correct biases in CRU and NASPA because it is highly accurate and spatially well-distributed with high-resolution."

**- Section 2.1. I recommend complementing this section with a description of why these specific datasets (excluding, of course, NASPA) were chosen. Such rationale is provided only for GPCC.**

We have added the sentences below for each data.

For CRU (lines 143 – 144): The CRU dataset was used because it is a well validated dataset that provides a long temporal coverage based on ground stations.

For GridMET (lines 147 – 149) : In this study, we assume the GridMET as a "ground truth" and use it to correct biases in CRU and NASPA because the GridMET incorporates satellite data, making it highly accurate and spatially well-distributed with high resolution.

**- Line 195-197. What does this mean?**

The hierarchical grouping in the GAM model structure allows for the separation of common trends and distinctive trends in each group. This structure is applied to our model to distinguish the long-term trend, which is common to all datasets from the seasonality that varies across each dataset. We have revised the sentence for clarification (Lines 231-234).

Original: When applied to our model, this involves a smoothed long-term trend that is common to all datasets (CRU, GridMET, and NASPA) and differences for the seasonal terms that repeat annually at the group-level (climate dataset) (Pedersen et al., 2019).

Revised: When applied, this model decomposes information from all datasets (CRU, GridMET, and NASPA) into a smoothed long-term trend that is common to all datasets and also an annual seasonality that varies slightly by dataset to account for bias relative to GridMET

**- Line 204. B or \Beta? This notation is not consistent with Equation 1 and 2.**

Fixed Line 204, and Line 227.

**- Line 234. How is the R2 calculated (e.g., calibration or cross-validation)?**

Stahle et al., (2020) have computed the coefficient of multiple determination (R2 in our study) during the calibration interval 1928–1978 compared to the GPCC dataset. This is separate from our study, but we have added this information to our text for context (Line 294 - 297).

NASPA reconstruction skills are investigated via calibration and validation statistics by data creators (Stahle et al., 2020). One of the calibration statistics, the coefficient of multiple determination ($R^2$) are presented in table 1.

**- Line 250. This statement should be supported by an analysis carried out at all locations—Figure 3 does not provide a representative sample.**

Thanks for the suggestion. An additional analysis was conducted to demonstrate that dsNASPA agrees well with GPCC across all locations, particularly in capturing ality. The Supplement Figure S1 illustrates all precipitation points at each season of 10 nearest neighbors, mean of the neighbors (dsNASPA), and GPCC data. It is evident that the dsNASPA and GPCC match across all years and capture seasonality. Please refer to Supplement Fig. S1. and we added reference in Line 316, which is the original statement: "The dsNASPA generally agrees with the GPCC, especially in capturing seasonality (Fig. 3 and Fig. S1)."

**- Line 255-256. I would include plots / results (perhaps in the SI if space is a constraints).**

To support the our statement regarding large nMAEs of dsNASPA in Los, CA,Phy, AZ, and Roe, NM, we have created nMAE plots (Supplemental Figure S2), and time series and scatter plots (Supplemental Figure S3) for all months and regions including the three aforementioned regions. The figures support arguments mentioned in main text, that: 1) The dsNASPA in most regions generally matches well with GPCC though seasonal differences in downscaling skills exist as expected: low skills in ASO season and the best skills in MJJ season, and 2) Large nMAE values are observed in low precipitation regions and seasons, specifically during warm season in Los, CA, Phx, AZ and Roe, NM.

Please refer to Supplement Fig. S3 and S4, and main text in Lines 325 where we have made the following change:

The Figures in S3 illustrate a few large precipitations in these regions drive large nMAE (scatter plot), however, dsNASPA still well matches with GPCC (time series).

**- Line 276. I would add to the SI the results concerning the other locations.**

Thanks for the comments. The bias captured in terms of modeled mean precipitation and drought (SPI = -1.5) and pluvials (SPI = +1.5, Figure 5) are presented for all locations in Supplement (Figure S4). We have added the following sentence in the manuscript in lines 370 - 376:

We present the results for all other regions in Fig. S4. The results indicate that the shape biases are largely dependent on the season, whereas mean biases are more dependent on the gauge. Notably, the ASO season shows large biases in the shape parameter. This is primarily because the dsNASPA in this season can't represent occasional extreme precipitation values, inducing an underestimation of its variance. In contrast, the MJJ season shows considerably less bias since the dsNASPA was developed from complete precipitation estimates in NASPA. The modeled mean parameters generally align well in most regions. A few exceptions exist in Mtv, CO and Grd, MT, where have large biases in the mean parameters across all seasons, possibly due to topographic effects between the gauge locations in these mountainous regions.

**- Line 290-292. This sentence should be supported by adequate plots.**

Figure S4 also shows the shape parameter biases as the modeled precipitation at drought (-1.5) and pluvials (+1.5) are determined by associated mean and shape parameters. Figure S4 indicates that all study sites have the typical pattern in shape parameter bias of dsNASPA shown in the Monte Vista, CO gauges (Figure 5), with largest bias in ASO and little bias in MJJ in all study sites as mentioned in the main text. The text added in previous comments (lines 370 -376) demonstrate the added plots.

Shape parameter bias results for Monte Vista, CO is typical of other gauges studied here, with largest bias during the interpolated ASO period and little bias in periods with good NASPA information.

**- Line 316. I don't see a "much larger" change in variance.**

Fixed as below (Lines 402 - 404):

Original: The variance between SPI = 1.5 and SPI = -1.5 during the MJJ season became much larger during the recent period, particularly for the ASO season, implying both wet and dry conditions have become more extreme in both directions than during prior centuries.

Revised: The variance between SPI = 1.5 and SPI = -1.5 became larger during the recent period, particularly for the ASO season, implying both wet and dry anomalies have become more extreme than during prior centuries.

**- Line 317-18. Is this conclusion really supported by data?**

Yes, it is. Figure 5 demonstrates the mentioned phenomenon. As mentioned, the upper limit of shaded line represents pluvial extremes (SPI = +1.5) and the lower limit of shaded line represents drought extreme (SPI = -1.5). Thus, larger variance between those extremes in ASO season implies the intensification of both extremes. Our result highlights the intensification of both extremes compared to the prior centuries.

**- Line 318. Is the trend statistically significant?**

Thanks for the comment. We deleted the word significant and just referred to "The drying trend" instead of "The significant drying trend".

**- Line 340. I would be careful with such a statement, since the analysis is based on 14 locations.**

We revised as follows (Line 431 – 433):

Original: Figure 7 implies a general spatial pattern of recent drying in the western US and wetting in the eastern US.

Revised: Figure 7 shows that the 14 demonstration sites generally follow a spatial climate pattern found in previous studies: with industrial era drying trends in the southwestern US, and wetting trends in eastern US (Lehner et al., 2018; Prein et al., 2016; Ellis and Marston, 2020).

**- Discussion. Would it be possible to extend this method to an entire gridded dataset? Which modelling challenges would be entailed by the problem of accounting for the spatial covariance of rainfall? Perhaps, the authors may want to elaborate on this point.**

Thank you for your comment. The authors have already applied our model to the entire extent of NASPA, covering southern Canada, the USA, and Mexico in a forthcoming paper under review. As you can see here, describing the model required a quite long paper, so we chose to separate the model derivation here from the wider application. In this other paper, currently under review, we have focused on dry and wet anomalies to investigate the long-term changes in extreme climates. In addition, the Global Climate Model (GCM) datasets were additionally incorporated including PMIP, Historical, and CMIP scenarios to enhance the credibility of the shared modeled trends and investigate future drought trends to 2100 years. The results are available at Sung et al (2022), DOI: 10.22541/essoar.168286969.91535356/v1.

Our model did not analyze spatial covariance; instead, we independently modeled each grid cell. However, we observed the spatial agreement in the modeled trends in neighboring regions, which provides the credibility of overall spatial trends found in that study.

(Kyungmin Sung, Gil Bohrer, James Howard Stagge. Centennial-scale intensification of wet and dry extremes in North America. Authorea. April 30, 2023.)

**References**

Ellis, A. W. and Marston, M. L.: Late 1990s' cool season climate shift in eastern North America, Climatic Change, 162, 1385–1398, https://doi.org/10.1007/s10584-020-02798-z, 2020.

Lehner, F., Deser, C., Simpson, I. R., and Terray, L.: Attributing the U.S. Southwest's Recent Shift Into Drier Conditions, Geophysical Research Letters, 45, 6251–6261, https://doi.org/10.1029/2018GL078312, 2018.

Prein, A. F., Holland, G. J., Rasmussen, R. M., Clark, M. P., and Tye, M. R.: Running dry: The U.S. Southwest's drift into a drier climate state, Geophysical Research Letters, 43, 1272–1279, https://doi.org/10.1002/2015GL066727, 2016.

Stahle, D. W., Cook, E. R., Burnette, D. J., Torbenson, M. C. A., Howard, I. M., Griffin, D., Diaz, J. V., Cook, B. I., Williams, A. P., Watson, E., Sauchyn, D. J., Pederson, N., Woodhouse, C. A., Pederson, G. T., Meko, D., Coulthard, B., and Crawford, C. J.: Dynamics, Variability, and Change in Seasonal Precipitation Reconstructions for North America, Journal of Climate, 33, 3173–3195, https://doi.org/10.1175/JCLI-D-19-0270.1, 2020.

---

## Referee Report (RR1)

Peer-review

"Assessing decadal to centennial scale nonstationary variability in meteorological drought trends"

General comments

This manuscript presents a clear description of the research gap, models, and analysis. The improvements in the document after the first peer-review round are substantial, providing a clear perspective on the research. Small corrections should be addressed to improve the manuscript.

Minor comments

Line 172. Please explain why the 94$^{th}$ percentile threshold and 6$^{th}$ percentile represent the precipitation associated with dry anomaly, and wet anomaly for each location?

Line 179. Please explain the insert K historical 13-Months precipitation sequences.

Line 312. This correspond to the K historical 13-months precipitation sequence? If this is the case should be explained better before.

---

## Author Response (AR2)

Peer-review
"Assessing decadal to centennial scale nonstationary variability in meteorological drought trends"
General comments

**Reviewer #1**
This manuscript presents a clear description of the research gap, models, and analysis. The improvements in the document after the first peer-review round are substantial, providing a clear perspective on the research. Small corrections should be addressed to improve the manuscript.

> The authors really appreciate your valuable comments to substantially improve the article. We did our best to respond to each of your helpful comments. Our responses are below.

Minor comments
Line 172. Please explain why the 94th percentile threshold and 6th percentile represent the precipitation associated with dry anomaly, and wet anomaly for each location?

> These percentiles correspond to SPI values of -1.5, 0, and 1.5 (see Line 150). These thresholds are commonly used in drought and pluvial studies to represent the precipitation associated with severe dry anomaly, typical, and wet conditions (Heim, 2002). For example, McKee et al. (1993) used -1.5 as a threshold between "Moderate" and "Severe" droughts. This is the original source for the SPI and the threshold has propagated into many later publications. The US Drought Monitor uses a similar threshold of -1.6.
>
> We have added a sentence to explain in the main text Line 150:
> These thresholds are commonly used in drought and pluvial studies to represent precipitation associated with dry anomaly, typical, and wet conditions (Heim, 2002), and a similar thresholds (5th percentile) is used by the US Drought Monitor (Svoboda et al. 2002).

Line 179. Please explain the insert K historical 13-Months precipitation sequences.

> We have added a sentence to explain 13months sequence.
> Lines 159 – 160:
> The 13-month sequence is considered a single downscaling unit containing three known NASPA values across a year (Figure 1).

Line 312. This correspond to the K historical 13-months precipitation sequence? If this is the case should be explained better before.

> Thank you for the suggestion. We made a clarification in lines 158-159. We believe this is smoothly connected to the detail explanation in lines 166-168.

Reviewer #2

The authors have addressed all my comments and I find the paper can be accepted for publication. Remaining comments (below) can be addressed in a final minor revision.

> The authors greatly appreciate for your time and thorough comments. We believe our manuscript has benefited from your helpful comments. Our responses are below.

L212: I think Gamma distribution parameters are not mean and shape, but shape and scale (pls revise throughout the text)

> The GAM model used by the 'mgcv' package estimates mean and shape parameters for the gamma distribution, instead of the shape and scale. The scale parameter can be then estimated as scale = mean/shape. We have modified equations 1 and 2 to clarify this.

> Estimating mean rather than scale is because the underlying model began as a more traditional gamma regression Generalized Linear Model (GLM), where the mean is estimated and the shape (or dispersion) parameter is estimated, but fixed. The mgcv package expands this to allow regression for the shape parameter. In addition to being more easily interpretable, a focus on mean and shape makes the resulting parameter estimates more robust.

> We have also added following statement in Lines 211 - 213:

> The Gamma distribution is typically prescribed by shape and scale parameters ($\alpha$ and $\theta$, respectively), but our approach follows Wood (2006), instead estimating the mean and shape parameters (Eqs. 2-3). The scale parameter can then be estimated from the mean and shape (Eq. 1).

L281: I do not see a generally good downscaling skill in season DJF in Fig. 4 (where there is also no tree growth in most climates), but in JFM-JJA. In addition, some stations show a good downscaling skill also in winter – can it be explained by the climate?

> Reconstruction skill in the original NASPA is often better during winter/spring (DJFMA) precipitation than summer (MJJ) for the southwestern US (Table 1 and Stahle et al. 2020). For example, Los Angeles, CA, Rodeo, NM, and Phoenix, AZ. This is because winters are generally warmer, the DJFMA period includes early spring, and winter rainfall generally sets up soil moisture for tree growth later in the year. Summers tend to be quite arid, making discriminating between precipitation anomalies more difficult for tree-rings, which has a warmer winter climate and often quite generally shows good skills in winter (Stahle et al., 2020).

> We have modified Lines 291- 298 to better clarify this:

> There are a few exceptions showing the best skill during winter (DJF) and poor skill, large nMAE, during early summer (MJJ, Figure 4). This occurs in only in the southwestern US (e.g. Los, CA, Phx, AZ and Roe, NM) where the underlying NASPA shows better initial reconstruction skill during their relatively mild winters (Table 1) and less skill in summer. For the dsNASPA, the seemingly large errors in nMAE during MJJ are primarily due to extremely small values in the denominator of the nMAE due to very low precipitation, combined with not capturing infrequent large rainfall events.

Fig. 5 Capture: Think there are no 'mean parameter' estimates shown. Rather estimates of the long-term trend in each data set.

> We have modified the figure caption as follows:

> Original: Mean parameter estimates for each dataset are represented…

> Fixed: The modeled long-term trends from each dataset are represented…

L321: Bias in the mean parameter? (Gamma has no mean parameter, see above). Check also for shape parameter…

> As mentioned above, the GAM model in 'mgcv' package directly fits the mean and shape parameters, whereas the scale parameter is a secondary estimate calculated from these two. We would like to leave the text as it is for the consistency, as the section 3.2 describes our model bias in technical context and mentions mean bias a few more times. However, as mentioned above, we have clarified this relationship between mean, shape, and scale in Eqs. 1-3, along with modifications to the text.

L344: (add at end of sentence): "… while the precipitation series are shown as raw data without bias correction for context."

> Added at the end of the sentence.

> Line 351:

> The solid black line shows the common, long-term trend of the mean while the precipitation series are shown as raw data without bias correction for context.

L 346: 'abrupt' appears too strong for what I would observe in Fig. 6

> We modified 'abrupt to 'noticeable' (line 353 now). We have also changed 'an abrupt drying trend' in line 351 to 'a drying trend'.

Fig- 6: "The yellow shaded area represents the 95% percentile for the Gamma distribution" should be reformulated. Think this is a 95% confidence interval of the long-term trend based on a Gamma distribution (or: assuming Gamma-distributed precipitation series)

> Agreed. It was a precipitation range between SPI = 1.5 and -1.5. We have modified the caption as below:

> The yellow shaded area represents the precipitation amount between SPI = +1.5 (upper boundary) and -1.5 (lower boundary) in the fitted Gamma distribution.

L359: Either "range between SPI = 1.5 and SPI = -1.5 …" Or "precipitation variance"

> Agreed. We changed to "The range between SPI = 1.5 and SPI = -1.5".

L. 385: One can hardly see this interpretation in the plots

> Agreed. We have removed the sentence to avoid confusion.

Supplement: Check figure captions

> We corrected the figure number.